# Nedd4 E3 ligase and beta-arrestins regulate ubiquitination, trafficking, and stability of the mGlu7 receptor

**Sanghyeon Lee[1,2], Sunha Park[1,2], Hyojin Lee[1,2], Seulki Han[1,2], Jae-man Song[1,2], Dohyun Han[3], Young Ho Suh[1,2]\***

[1]Department of Biomedical Sciences, Seoul National University College of Medicine, Seoul, Republic of Korea; [2]Neuroscience Research Institute, Seoul National University College of Medicine, Seoul, Republic of Korea; [3]Proteomics Core Facility, Biomedical Research Institute, Seoul National University Hospital, Seoul, Republic of Korea

**Abstract** The metabotropic glutamate receptor 7 (mGlu7) is a class C G protein-coupled receptor that modulates excitatory neurotransmitter release at the presynaptic active zone. Although post-translational modification of cellular proteins with ubiquitin is a key molecular mechanism governing protein degradation and function, mGlu7 ubiquitination and its functional consequences have not been elucidated yet. Here, we report that Nedd4 ubiquitin E3 ligase and β-arrestins regulate ubiquitination of mGlu7 in heterologous cells and rat neurons. Upon agonist stimulation, β-arrestins recruit Nedd4 to mGlu7 and facilitate Nedd4-mediated ubiquitination of mGlu7. Nedd4 and β-arrestins regulate constitutive and agonist-induced endocytosis of mGlu7 and are required for mGlu7-dependent MAPK signaling in neurons. In addition, Nedd4-mediated ubiquitination results in the degradation of mGlu7 by both the ubiquitin-proteasome system and the lysosomal degradation pathway. These findings provide a model in which Nedd4 and β-arrestin act together as a complex to regulate mGlu7 surface expression and function at presynaptic terminals.

DOI: https://doi.org/10.7554/eLife.44502.001

**\*For correspondence:**
suhyho@snu.ac.kr

**Competing interests:** The authors declare that no competing interests exist.

## Introduction

The metabotropic glutamate receptor 7 (mGlu7) is a class C G protein-coupled receptor (GPCR) that regulates glutamate neurotransmitter release at presynaptic terminals via inhibitory heterotrimeric G-protein signaling cascades. mGlu7 is only activated under conditions of sustained synaptic activity due to its low affinity for glutamate, thereby acting as an auto-inhibitory receptor (*Niswender and Conn, 2010*). Like many GPCRs, the function and desensitization of mGlu7 are tightly regulated by mechanisms of receptor endocytosis from the plasma membrane. In particular, agonist-induced mGlu7 endocytosis underlies a presynaptic form of long-term depression (LTD) and regulates a metaplastic switch at the mossy fiber-CA3 stratum lucidum interneuron (MF-SLIN) synapses. At naive MF-SLIN synapses, high frequency stimulation (HFS) or agonist L-AP4 treatment triggers presynaptic LTD by mGlu7 activation. Following L-AP4-induced internalization of mGlu7, the same HFS produces presynaptic LTP, which requires the activation of the adenylate cyclase-cAMP-PKA pathway and subsequent action of RIM1α (*Pelkey et al., 2005*; *Pelkey et al., 2008*; *Suh et al., 2008*).

A number of cellular processes related to GPCR trafficking, signaling, stability, and agonist sensitivity are regulated by ubiquitination, a post-translational modification (PTM) (*Alonso and Friedman, 2013*; *Dores and Trejo, 2012*; *Marchese et al., 2008*; *Sarker et al., 2011*). Ubiquitination of target substrates is achieved by the covalent attachment of ubiquitin to lysine residues of substrates

through a three-enzyme cascade catalyzed by E1 (ubiquitin activating), E2 (ubiquitin conjugating), and E3 (ubiquitin ligase) enzymes. Whereas misfolded GPCRs during de novo synthesis in the ER are poly-ubiquitinated and degraded via an ER-associated degradation pathway, certain GPCRs in the plasma membrane are ubiquitinated at a steady state by specific E3 ubiquitin ligases (*Dores and Trejo, 2012*). The E3 ubiquitin ligase catalyzes the transfer of the ubiquitin moiety from E2 enzymes to the target substrates and plays a key role in determining the substrate specificity of ubiquitination. Among the several hundred E3 ubiquitin ligases in mammals, Nedd4 (Neural precursor cell-expressed developmentally down-regulated 4) (also known as Nedd4-1) E3 ligase, which belongs to the HECT (Homologous to E6-associated protein Carboxyl Terminus) domain-containing family (*Rotin and Kumar, 2009*), has been well characterized in its regulation of the turnover and trafficking of ion channels and GPCRs present in neurons such as voltage-gated sodium channels, calcium channels, GluA1 AMPA receptor, GluN2D-containing NMDA receptor, $Na^+/H^+$ exchanger 1, and β2-adrenergic receptor (β2AR) (*Boase and Kumar, 2015*; *Gautam et al., 2013*; *Han et al., 2013*; *Lin et al., 2011*; *Rougier et al., 2011*; *Schwarz et al., 2010*; *Shenoy and Lefkowitz, 2011*; *Shenoy et al., 2008*; *Simonin and Fuster, 2010*).

β-arrestin 1 (also known as arrestin 2) and β-arrestin 2 (also known as arrestin 3) are cytosolic adaptor proteins involved in the regulation of GPCR desensitization, trafficking, and signaling (*DeWire et al., 2007*; *Luttrell and Gesty-Palmer, 2010*). β2AR is one of the best-characterized GPCRs with regard to functional consequences of GPCR-β-arrestin interaction. Following agonist stimulation, β-arrestins are rapidly recruited to phosphorylated activated β2AR at the plasma membrane. β-arrestins typically terminate the interaction of receptors with G-proteins, however recent studies indicate that β-arrestins trigger the ubiquitination of receptors and give rise to distinct signaling pathways independently of G-proteins. Upon agonist stimulation of β2AR, β-arrestins act as adaptors to initiate recruitment of E3 ubiquitin ligases such as Nedd4 or Mdm2 as well as the clathrin-AP2 adaptor complex, thereby promoting endocytosis of β-arrestin-bound receptors (*Jean-Charles et al., 2016*; *Shenoy et al., 2001*; *Shenoy et al., 2008*). Once internalized to early endosomes, receptors can either be recycled back to the plasma membrane or designated to degradation pathways (*Marchese et al., 2008*). Agonist-stimulated GPCRs associated with β-arrestins can generate two stages of signaling. While class A receptors (e.g. β2AR) exhibit a transient interaction with β-arrestins, undergo recycling pathways, and induce weak extracellular signal-regulated kinase (ERK) signaling, class B receptors such as Angiotensin II type 1a receptor ($AT_{1a}R$) and vasopressin V2 receptor (V2R) display more robust interaction with β-arrestins that gives rise to the receptor-G protein-β-arrestin super-complex in the endosomes, induce robust and sustained ERK activity, and are targeted for degradation (*Eichel et al., 2018*; *Ranjan et al., 2017*; *Shenoy and Lefkowitz, 2011*; *Thomsen et al., 2016*). However, the roles of β-arrestins in regulating trafficking and function of class C GPCRs remain poorly understood.

In the present study, we find that mGlu7, a class C GPCR is ubiquitinated by Nedd4 E3 ligase in heterologous cells and cultured neurons. We show that Nedd4 and β-arrestins are constitutively associated with mGlu7 and recruited to mGlu7 by agonist stimulation, revealing the essential roles of β-arrestin as an adaptor that recruits Nedd4 to mediate ubiquitination and endocytosis of mGlu7. In addition, we demonstrate that β-arrestin 2 and Nedd4 regulate mGlu7-mediated mitogen-activated protein kinase (MAPK) signaling, and Nedd4 is required for agonist-induced receptor degradation via both the proteasomal and the lysosomal pathways. These findings reveal the roles of the Nedd4-β-arrestin complex in regulating ubiquitination and surface stability of mGlu7, and each of their functional effects at presynaptic terminals.

## Results

### mGlu7 is ubiquitinated by agonist stimulus or Nedd4 E3 ligase in heterologous cells and neurons

To explore if ubiquitination determines mGlu7 trafficking and function, we first tested if mGlu7 could be modified by ubiquitination in heterologous cells. We co-transfected c-myc epitope-tagged mGlu7a in the extracellular N-terminal domain (myc-mGlu7) and hemagglutinin epitope-tagged ubiquitin (HA-Ub) in HEK 293 T cells. The transfected cell lysates were immunoprecipitated with anti-myc antibody and blotted with anti-HA antibody to detect the ubiquitin-conjugated mGlu7. We

observed diffuse high-molecular-weight mGlu7 immunoreactivities in the lane co-transfected with mGlu7 and ubiquitin, but not in the control lane transfected with either mGlu7 or ubiquitin, indicative of ubiquitinated mGlu7 signals (*Figure 1A*). We found that the ubiquitinated mGlu7 bands are present mainly in dimeric and multimeric forms with apparent molecular weights larger than 200 kDa, whereas the monomeric form of mGlu7 (~100 kDa) is barely ubiquitinated (*Figure 1A*).

To explore if agonist-induced activation of mGlu7 induces receptor ubiquitination, we treated HEK 293 T cells with 1 mM L-glutamate (L-Glu) following co-expression of myc-mGlu7 and HA-Ub. We were able to observe an approximately 2-fold increase in mGlu7 ubiquitination within 10 min of agonist stimulation, which was sustained up to 60 min. Total mGlu7 level was reduced after 30 min most likely due to receptor degradation (*Figure 1B*). Next, we examined the endogenous ubiquitination of mGlu7 in cultured cortical neurons following Group III mGlu receptors agonist L-AP4 (400 µM) treatment for 5 min. The ubiquitinated mGlu7 was pulled down using an anti-ubiquitin antibody (clone FK2) and blotted with mGlu7 antibody. We found that L-AP4 treatment increases endogenous mGlu7 ubiquitination in neurons by approximately 2-fold (*Figure 1C*). Conversely, we detected ubiquitinated bands of endogenous mGlu7 after immunoprecipitation with anti-mGlu7 antibody in neurons (*Figure 1—figure supplement 1*). To distinguish whether surface-expressed or intracellular mGlu7 undergoes ubiquitin modification, we devised a two-round purification strategy after cell surface biotinylation and separated the surface versus internal ubiquitinated proteins (*Figure 1D*). Specifically, cultured cortical neurons were labeled by membrane-impermeable biotin and then incubated with L-AP4 for 5 min. Cell lysates were then pulled down with Streptavidin-agarose beads. The supernatant was considered to be the intracellular fraction, and the surface fraction was obtained from bead-bound proteins. The ubiquitinated proteins were subsequently isolated from the surface or intracellular fraction using FK2 antibodies, respectively. We found that surface-expressed mGlu7 undergoes agonist-induced ubiquitination, whereas intracellular mGlu7 does not (*Figure 1D*). This result suggests that mGlu7 ubiquitination occurs in combination with an agonist-induced endocytosis process.

As E3 Ub ligase largely determines the substrate specificity of ubiquitination, we searched for the E3 ligases responsible for mGlu7 ubiquitination. We tested a Nedd4 E3 ligase because Nedd4 has been implicated in regulating the turnover and trafficking of many ion channels and GPCRs in neurons (*Han et al., 2013*; *Lin et al., 2011*; *Rougier et al., 2011*; *Schwarz et al., 2010*; *Shenoy and Lefkowitz, 2011*; *Shenoy et al., 2008*; *Simonin and Fuster, 2010*). When Nedd4 wild type (WT) was co-expressed in HEK 293 T cells, we observed a robust ubiquitination of mGlu7, whereas Nedd4 C867S, a catalytically inactive form of Nedd4 (*Blot et al., 2004*; *Kamynina et al., 2001*), did not alter the ubiquitination level of mGlu7 (*Figure 1E*). To confirm the essential role of Nedd4 in agonist-induced mGlu7 ubiquitination, we knocked down the endogenous expression of Nedd4 using lentivirus-mediated short hairpin RNA (shRNA) in cultured cortical neurons. Nedd4 shRNA was highly effective in knocking down endogenous Nedd4 expression 7 days after infection. Notably, agonist L-AP4-induced mGlu7 ubiquitination was markedly reduced by Nedd4 knockdown (Nedd4 KD), indicating that Nedd4 is required for agonist-induced mGlu7 ubiquitination (*Figure 1F*). To examine the interaction of Nedd4 with its substrate mGlu7, endogenous mGlu7 from cultured cortical neurons was immunoprecipitated and blotted with anti-Nedd4 antibody. We observed constitutive interaction between endogenous Nedd4 and mGlu7, and their interaction was markedly enhanced after treatment of L-AP4 for 5 min (*Figure 1G*). Taken together, these results indicate that Nedd4 is recruited to mGlu7 upon agonist-stimulation and plays a role as an E3 ubiquitin ligase for mGlu7 ubiquitination.

It is well known that among seven lysine residues of the ubiquitin protein, the formation of K48- or K63-linked polyubiquitin chain typically directs protein substrates to the proteasomal degradation or endocytic trafficking, respectively (*Alonso and Friedman, 2013*; *Dores and Trejo, 2012*). To identify which types of ubiquitin linkages are preferentially attached to mGlu7, HA-ubiquitin K48 (ubiquitin with only a K48 residue, other lysines mutated to arginines) or HA-K63 (ubiquitin with only a K63 residue, other lysines mutated to arginines) expression plasmids were co-transfected with mGlu7. Although K63 ubiquitin was more efficiently conjugated to mGlu7 than K48 ubiquitin, there was no significant difference between K48- or K63-linked mGlu7 ubiquitination when their mGlu7 ubiquitination levels were normalized by the total expression levels of ubiquitin K48 or K63, respectively (*Figure 1H*). This result implicates that ubiquitination of mGlu7 may regulate receptor endocytosis as well as degradation.

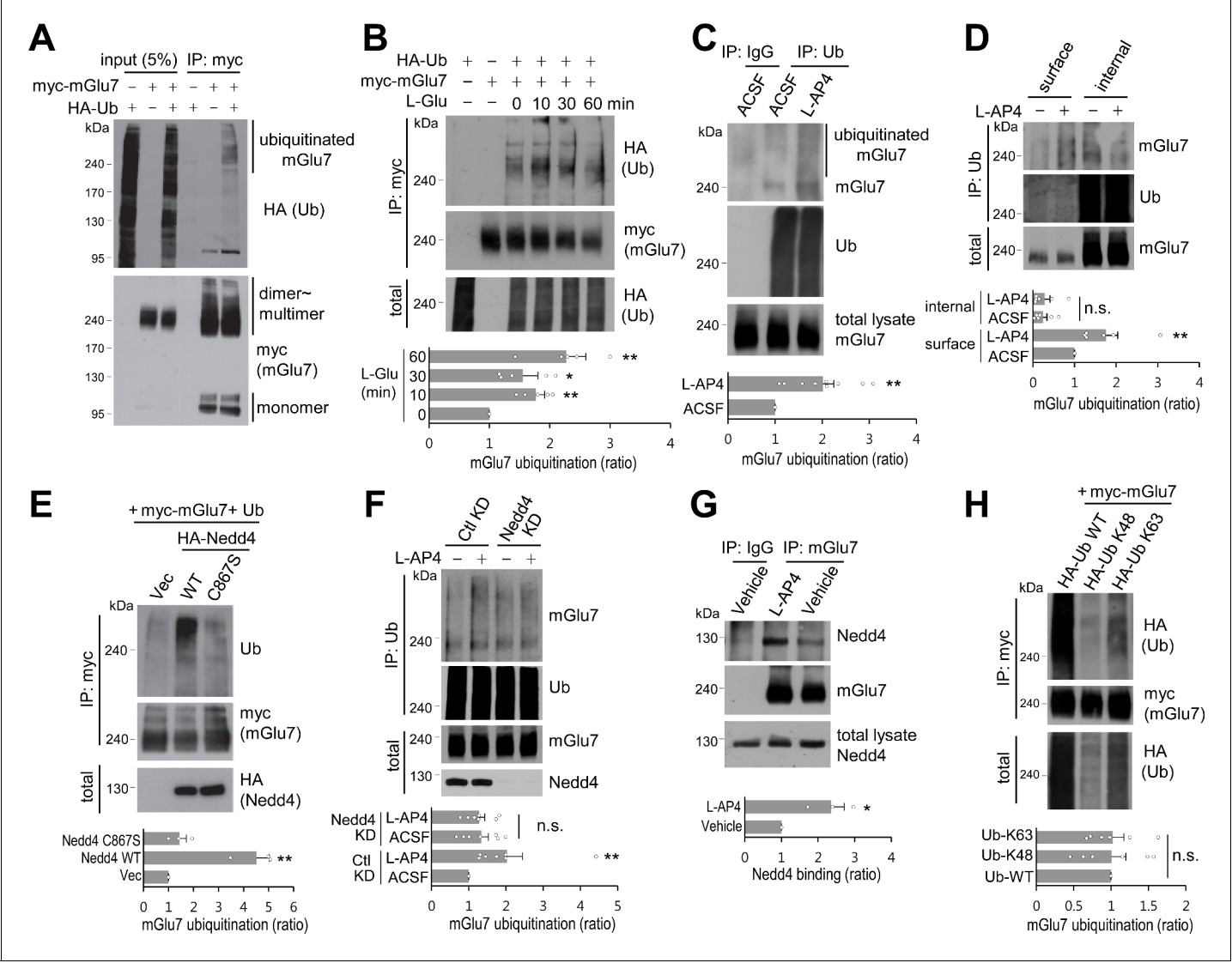

**Figure 1.** mGlu7 is ubiquitinated by agonist treatment or Nedd4 expression in heterologous cells and neurons. (**A**) mGlu7 is ubiquitinated in HEK 293 T cells. N-terminal c-myc epitope-tagged mGlu7a (myc-mGlu7) was co-transfected with HA epitope-tagged ubiquitin (HA-Ub) in HEK 293 T cells. Cell lysates were immunoprecipitated with anti-myc antibody (9E10) and immunoblotted with anti-HA antibody. Diffuse bands of high molecular weight larger than 200 kDa representing ubiquitinated mGlu7 were detected in the lane that the co-expressed mGlu7 and Ub were loaded in. (**B**) HA-Ub and myc-mGlu7 were co-transfected in HEK 293 T cells. Agonist L-Glutamate (L-Glu, 1 mM) was added for 10 to 60 min before cell lysis. Bar graph below represents L-Glu-induced ubiquitination levels normalized to the 0 min timepoint sample (means ± SEM; 10 min, 1.77 ± 0.15; 30 min, 1.55 ± 0.25; 60 min, 2.27 ± 0.33; n = 5, *p<0.05, **p<0.01 versus 0 min, one-way ANOVA). (**C**) Cultured cortical neurons at 14 days in vitro (DIV) were incubated in ACSF (140 mM NaCl, 5 mM KCl, 10 mM HEPES, 10 mM D-glucose, 2 mM CaCl$_2$, 2 mM MgCl$_2$ plus 10 µM MG132 and 52.5 µM leupeptin) for 10 min and then 400 µM L-AP4 was added for 5 min at 37°C. Endogenous mGlu7 ubiquitination in neurons was detected by immunoprecipitation using anti-ubiquitin antibody (FK2) and western blotting using anti-ubiquitin (P4D1) or anti-mGlu7 antibody. Bar graph below represents mean ± SEM (L-AP4, 2.02 ± 0.23; n = 9, **p<0.01, Student's t-test). (**D**) After cell surface biotinylation with membrane impermeable Sulfo-NHS-SS-Biotin, cortical neurons were treated with L-AP4 for 5 min. The surface proteins were isolated by Streptavidin-agarose beads overnight at 4°C and eluted by incubating the beads with 50 mM DTT for 30 min at 50°C. The eluted proteins (surface) and unbound lysates (internal) were further immunoprecipitated by anti-ubiquitin antibody (FK2), and western blotted with the indicated antibodies. Bar graph below represents means ± SEM of relative mGlu7 ubiquitination levels normalized to total mGlu7 and are shown as a ratio to the non-stimulated control of the surface fraction (surface, L-AP4, 1.75 ± 0.28; internal, ACSF, 0.25 ± 0.09; internal, L-AP4, 0.28 ± 0.13; n = 6, **p<0.01, one-way ANOVA). (**E**) Nedd4 E3 ligase regulates agonist-stimulated ubiquitination of mGlu7. HA-tagged Nedd4 WT or C867S mutant was co-expressed with Ub and myc-mGlu7 in HEK 293 T cells. Ubiquitination of mGlu7 was analyzed as above. Bar graph below represents mean ± SEM of Nedd4-induced mGlu7 ubiquitination levels normalized to the vector control (Nedd4 WT, 4.51 ± 0.52; Nedd4 C867S, 1.44 ± 0.27; n = 3, **p<0.01, one-way ANOVA). (**F**) Cultured cortical neurons were infected with lentiviruses harboring Nedd4 shRNA (Nedd4 KD) or non-related control shRNA (Ctl KD) for 7 days. At DIV 14, L-AP4-induced endogenous mGlu7 ubiquitination was evaluated as in **Figure 1C**. Bar graph

*Figure 1 continued on next page*

*Figure 1 continued*

represents means ± SEM (Ctl KD + L-AP4, 2.03 ± 0.41; Nedd4 KD + ACSF, 1.33 ± 0.19; Nedd4 KD + L-AP4, 1.28 ± 0.14; n = 7, **p<0.01, n.s. indicates p>0.05, one-way ANOVA). (G) Analysis of endogenous interaction between Nedd4 E3 ligase and mGlu7. Cortical neurons at DIV 14 were treated with L-AP4 for 5 min, and cell lysates were immunoprecipitated with anti-mGlu7 antibody and western blotting was carried out with the indicated antibodies. Bar graph represents mean ± SEM (L-AP4, 2.36 ± 0.36; n = 3, *p<0.05, Student's *t*-test). (H) HA-tagged K48 or K63 Ub was co-transfected with mGlu7 in HEK 293 T cells. The cells were treated with MG132 (10 μM) and leupeptin (52.5 μM) for 5 hr to inhibit the proteasomal and lysosomal degradation of the receptor. The cells were stimulated with L-Glu for 5 min before harvesting. K48- or K63-mediated ubiquitination levels were quantified after normalization to total ubiquitination levels. Bar graph below represents mGlu7 band intensities normalized to Ub-WT (means ± SEM; Ub-K48, 1.01 ± 0.19; Ub-K63, 1.02 ± 0.15; n = 6, n.s. indicates p>0.05, one-way ANOVA).

DOI: https://doi.org/10.7554/eLife.44502.002

The following source data and figure supplement are available for figure 1:

**Source data 1.** Ubiquitination of mGlu7 in heterologous cells and neurons.
DOI: https://doi.org/10.7554/eLife.44502.004

**Figure supplement 1.** Endogenous mGlu7 is ubiquitinated in cortical neurons.
DOI: https://doi.org/10.7554/eLife.44502.003

## mGlu7 is ubiquitinated on the lysine residues of the C-terminus and intracellular loop 2

We hypothesized that ubiquitin-conjugation of mGlu7 occurs in the eight lysine residues in the C-terminal tail (CT) of mGlu7 (*Figure 2A*). To determine which lysine residues are responsible for ubiquitin-conjugation of mGlu7, we generated sequential deletion mutants (mGlu7 Δ893, Δ879, Δ860, and Δ857) by introducing a stop codon at the position of designated amino acid numbers, and examined ubiquitination of these mutants. Unexpectedly, the level of ubiquitination was not reduced in any deletion mutants including mGlu7 Δ857, which does not possess any lysine residues in its CT (*Figure 2B*). We recognized four additional lysine residues present in the intracellular loop (iL) 2 and 3 domains (*Figure 2A*), and thus generated iL mutants such that all four lysine residues in iLs were individually replaced with arginine residues (mGlu7 iL 4K4R) by site-directed mutagenesis. Consistent with the data using deletion mutants, the mGlu7 CT 8K8R mutant, in which all eight lysine residues in mGlu7 CT were mutated to arginines, was as efficiently ubiquitinated as mGlu7 WT after treatment of L-Glu for 5 min (*Figure 2C*). However, the efficiency of ubiquitination of mGlu7 iL 4K4R was also similar to that of mGlu7 WT or CT 8K8R (*Figure 2C*), suggesting that the lysines on iLs are not the only ubiquitination sites. The ubiquitination of the mGlu7 12K12R mutant in which all twelve lysine residues were mutated to arginines was almost completely abolished (*Figure 2C*). Taken together, these results suggest that mGlu7 is ubiquitinated at lysine residues of both iLs and CT.

To further map the specific lysine residues to which ubiquitin is covalently attached on the iL domains of mGlu7, lysine residues in the iL domains were mutated to arginines in combination with mGlu7 Δ857 mutant. We found that K688 and K689 mutations in the iL2 (iL K688/689R) with mGlu7 Δ857 lead to the marked reduction of mGlu7 ubiquitination to a similar level as iL 4K4R with mGlu7 Δ857, suggesting the K688 and K689 residues are ubiquitinated residues in the iL domains of mGlu7 (*Figure 2D*). However, in spite of further mutagenesis studies we were not able to identify dominant ubiquitination sites among eight mGlu7 CT resides (data not shown), suggesting that the two lysine residues K688 and K689 in iL2 and all eight lysine residues in CT are the target ubiquitination residues of mGlu7.

## β-arrestins are recruited to mGlu7 by agonist stimulation and facilitate ubiquitination of mGlu7

Previous studies have shown that β-arrestin 2 plays a role as an adaptor for β2AR ubiquitination by recruiting Nedd4 E3 ligase upon agonist stimulation (*Shenoy et al., 2008*). It has also been reported that β-arrestins are involved in the desensitization, endocytosis, and intracellular signaling of Group I mGlu receptors (reviewed in *Suh et al., 2018*). Thus we investigated the roles of β-arrestins in the function and trafficking of Group III mGlu receptor mGlu7. GFP-tagged β-arrestin 1 or 2 was co-transfected with FLAG-tagged mGlu7 or β2AR in HEK cells, and receptors were immunoprecipitated by anti-FLAG antibody. As evident from *Figure 3—figure supplement 1*, β-arrestin 1 robustly binds to mGlu7 as well as β2AR, whereas the binding affinity of β-arrestin 2 to mGlu7 appears to be lower (*Figure 3—figure supplement 1*). We hypothesized that agonist-stimulated recruitment of Nedd4

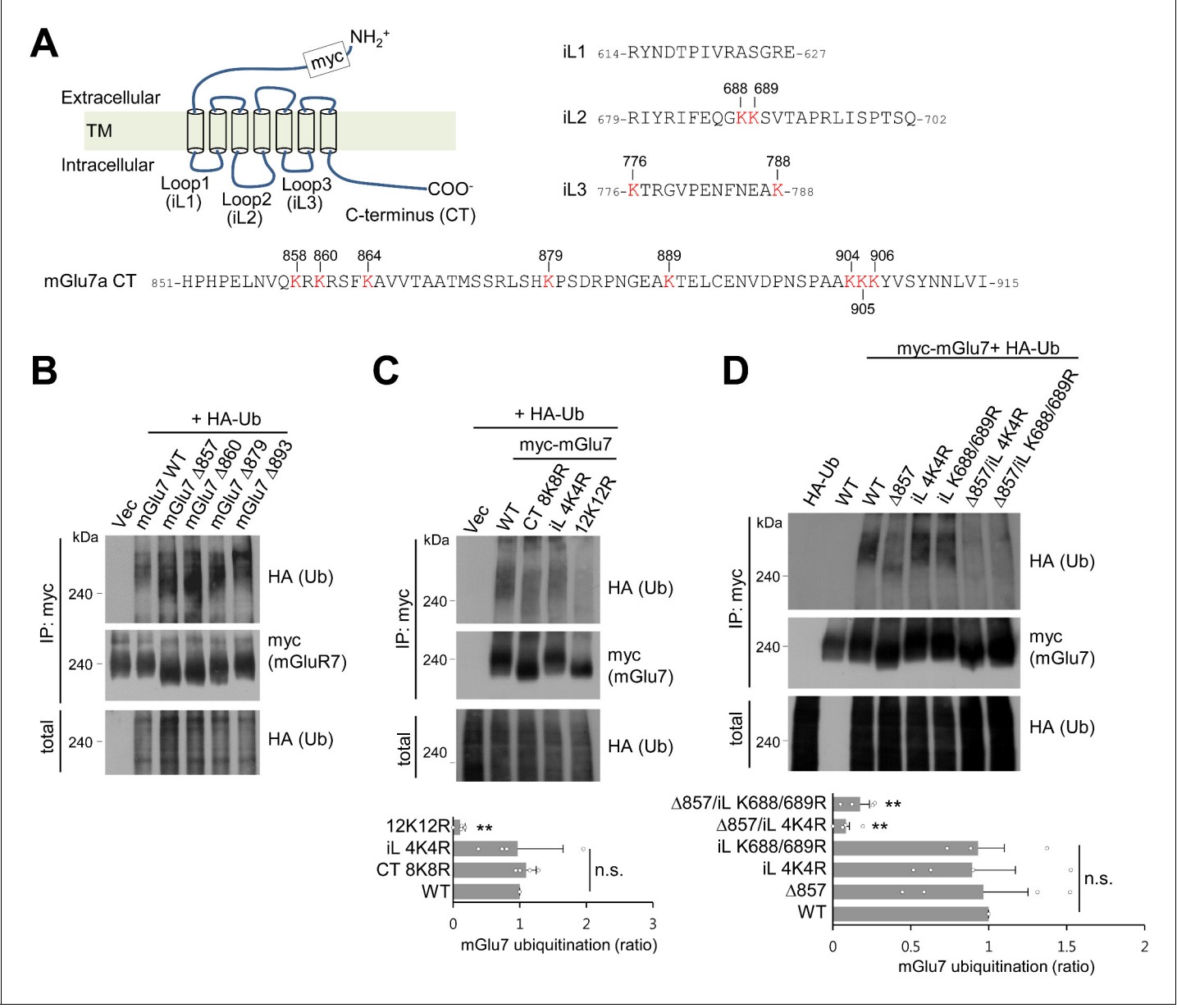

**Figure 2.** mGlu7 is ubiquitinated at lysine residues in both CT and iL2 domains. (**A**) A schematic diagram showing amino acid sequences of mGlu7 intracellular loop regions (iL1–3) and the cytoplasmic C-terminus tail (CT) of rat mGlu7a. The twelve lysine (K) residues in the CT and iL domains are displayed in red. (**B**) Myc-mGlu7 WT or sequential deletion mutants at the designated position were co-transfected with HA-Ub in HEK 293 T cells. Cell lysates were immunoprecipitated with anti-myc antibody and immunoblotted with anti-HA antibody. Diffuse bands of high molecular weight larger than 200 kDa represent ubiquitinated mGlu7. (**C**) Ubiquitination of mGlu7 WT or mutants in which the lysine (K) residues have been substituted to arginine (R) residues. Ubiquitination levels were analyzed as in *Figure 1A* after 5 min treatment of 1 mM L-Glu. CT 8K8R had all eight lysine residues in the mGlu7 CT mutated to arginines; iL 4K4R, all four lysine residues in the mGlu7 iL domains to arginines; 12K12R, all twelve lysine residues in the mGlu7 CT and iL domains to arginines. CT, C-terminal tail; iL, intracellular loop. Bar graph below represents mean ± SEM showing quantification of ubiquitination levels in the mutants normalized to WT (CT 8K8R, 1.09 ± 0.08; iL 4K4R, 0.97 ± 0.34; 12K12R, 0.11 ± 0.04; n = 4, **p<0.01, n.s. indicates p>0.05, one-way ANOVA). (**D**) Ubiquitination of mGlu7 in the intracellular loops in HEK 293 T cells. mGlu7 Δ857 does not harbor any lysine residues in the CT due to the stop codon at amino acid position 857. mGlu7 iL K688/689R represents two lysine residues in the iL2 are mutated to arginines. mGlu7 Δ857/iL 4K4R or mGlu7 Δ857/iL K688/689R are mutants that combine Δ857 and iL 4K4R or iL K688/689R mutations, respectively. Bar graph below represents mean ± SEM of band intensities normalized to WT (Δ857, 0.97 ± 0.29; iL 4K4R, 0.89 ± 0.28; iL K688/689R, 0.93 ± 0.17; Δ857/iL 4K4R, 0.08 ± 0.02; Δ857/iL K688/689R, 0.17 ± 0.06; n = 4, **p<0.01, n.s. indicates p>0.05, one-way ANOVA).

DOI: https://doi.org/10.7554/eLife.44502.005

The following source data is available for figure 2:

**Source data 1.** Mapping lysine residues of mGlu7 ubiquitination.
DOI: https://doi.org/10.7554/eLife.44502.006

to mGlu7 requires β-arrestins that play as adaptors to link mGlu7 and Nedd4. Accordingly, we characterized the time course of binding among β-arrestins, Nedd4, and mGlu7. FLAG-mGlu7, β-arrestin 1- or 2-GFP, and HA-Nedd4 were co-transfected in HEK 293 T cells and mGlu7 was immunoprecipitated by anti-FLAG antibody (*Figure 3A,B*). We observed that β-arrestins and Nedd4 bind constitutively to mGlu7 before stimulation with agonist L-Glu (*Figure 3A,B*). Of particular interest, agonist stimulation rapidly increased binding of β-arrestin 1 to mGlu7 within 5 min by approximately 1.8 fold relative to basal binding levels (*Figure 3A,C*). This interaction was sustained for 15 min and decreased to basal levels after 30 min (*Figure 3A,C*). Binding of Nedd4 to mGlu7 was also elevated after 5 min of agonist stimulation when β-arrestin 1 was co-expressed (*Figure 3A,D*). After 60 min of agonist treatment, binding of β-arrestin 1 or Nedd4 to mGlu7 was reduced, probably due to the degradation of the receptor (*Figure 3A,C,D*). Binding of β-arrestin 2 to mGlu7 exhibited a smaller degree of increase by 1.2 fold at 5 min after agonist treatment (*Figure 3B,C*). Nedd4 binding to mGlu7 was also increased at 5 min of agonist stimulation when β-arrestin 2 was co-expressed (*Figure 3B,D*). We also examined the time course of binding between Nedd4 and β-arrestin 1 when mGlu7 was activated. We found that β-arrestin 1 and Nedd4 are rapidly associated within 1 min and their interaction continued more than 10 min after mGlu7 agonist stimulation (*Figure 3E,F*), suggesting that Nedd4 is recruited to the activated mGlu7 by β-arrestin 1.

To investigate if endogenous mGlu7, Nedd4, and β-arrestin 1 are present at the same complex, we performed co-immunoprecipitation assay using anti-β-arrestin 1 antibody in cortical neurons. We have found that three proteins are present in the same complex and that their binding affinity is increased by agonist treatment (*Figure 3G,H*). However, binding affinity of Nedd4 or β-arrestin 1 to different mGlu7 ubiquitination site mutants was not altered in neurons (*Figure 3—figure supplement 2*). Next, to confirm the role of β-arrestins in the recruitment of Nedd4 to mGlu7, we knocked down β-arrestin 1 or 2 by co-transfection with β-arrestin 1 or 2 shRNA in HEK 293 T cells, respectively. We found that either β-arrestin 1 (β-arr1 KD) or 2 knockdown (β-arr2 KD) abolished an increase in agonist-induced binding of Nedd4 to mGlu7 (*Figure 3I,J*), To examine the requirement for β-arrestin 1 and 2 in Nedd4 recruitment to mGlu7 in neurons, we generated lentiviruses harboring β-arrestin 1 or 2 shRNA under H1 promoter. When β-arrestin 1 or 2 was knocked down in primary cortical neurons, agonist-induced Nedd4 recruitment to mGlu7 was not impaired. However, when both β-arrestin 1 and 2 were simultaneously knocked down, we observed a significant reduction of agonist-induced Nedd4 recruitment to mGlu7 (*Figure 3K,L*, *Figure 3—figure supplement 3*). Endogenously expressed levels of β-arrestins are high in the brain, but relatively low in HEK 293 T cells because only mGlu7 and Nedd4 were exogenously overexpressed in HEK 293 T cells (*Attramadal et al., 1992*; *Barlic et al., 1999*). Therefore, we hypothesize that β-arrestin 1 and 2 may compensate for the function of each other in neurons due to their high expression but not in heterologous cells due to their stoichiometrically low expression. Taken together, these results indicate that agonist-induced recruitment of Nedd4 to mGlu7 is mediated by β-arrestins.

Our results indicate that β-arrestins and Nedd4 form a complex with mGlu7 in the basal state, and upon agonist stimulation β-arrestins are further recruited to mGlu7, which promotes Nedd4 binding to mGlu7. Therefore we tested whether β-arrestins affect mGlu7 ubiquitination by recruiting Nedd4 in cultured cortical neurons. We found that agonist-induced mGlu7 ubiquitination levels were significantly reduced by either β-arrestin 1 or 2 KD, and further decreased by double KD of β-arrestin 1 and 2 compared with the control non-related target knockdown (Ctl KD) in cultured cortical neurons (*Figure 3M,N*, *Figure 3—figure supplement 4*). These results indicate that β-arrestins recruit Nedd4 to mGlu7 and facilitate mGlu7 ubiquitination upon agonist stimulation, and agonist-induced mGlu7 ubiquitination is regulated by the recruitment of β-arrestins.

## Nedd4 and β-arrestins bind to the CT and iL2 of mGlu7

To determine the binding domains between Nedd4 and mGlu7, we performed GST pull-down assays using GST-fused mGlu7 CT or iL domains which were immobilized on Glutathione-Sepharose 4B beads. Specifically, GST-fused mGlu7 iL1, iL2, iL3, mGlu7a CT, or mGlu7b CT domains were incubated with lysates from the whole rat brains and the binding proteins to each domain were isolated. mGlu7b is a splice variant of mGlu7 and has different amino acid sequences only in its distal CT

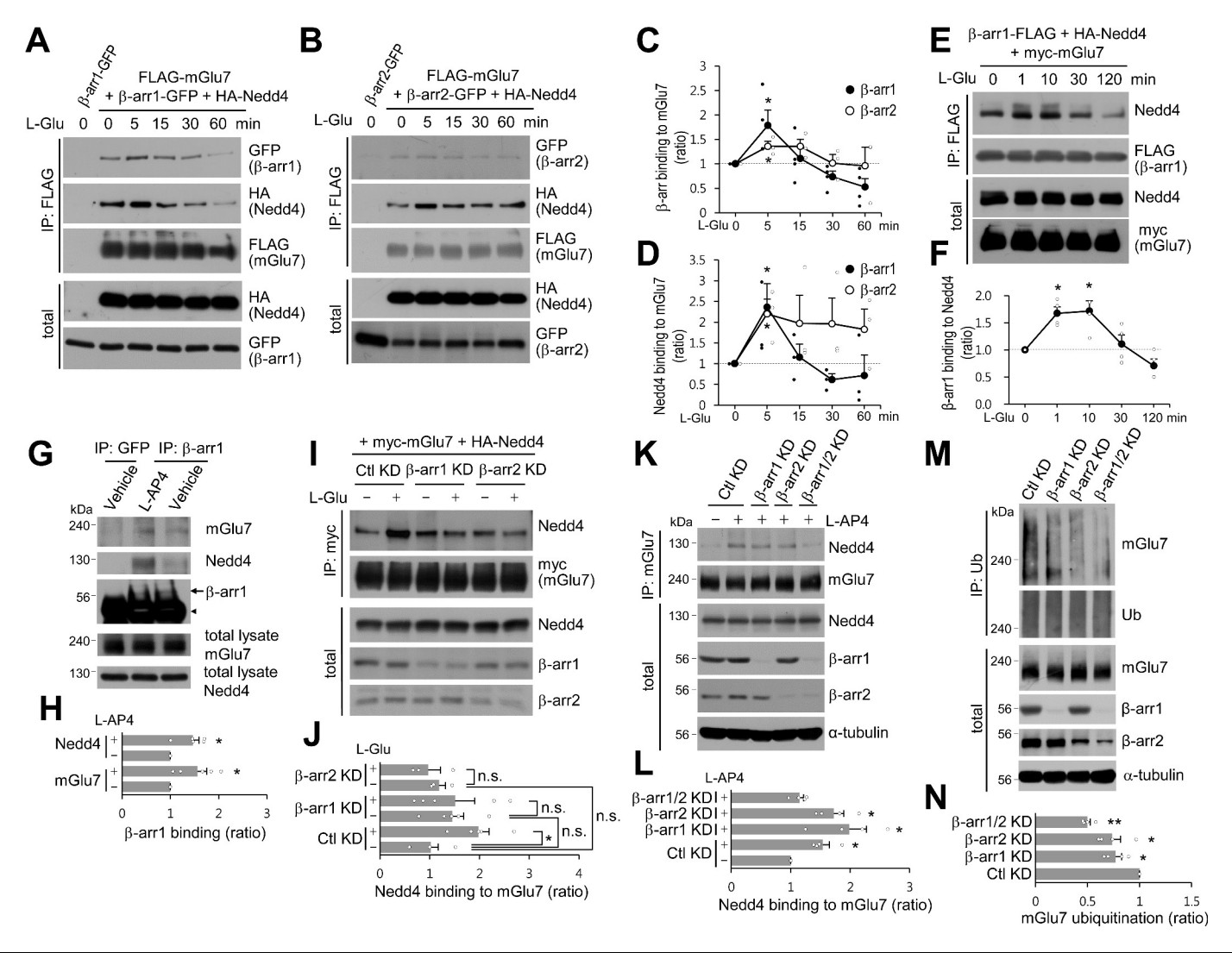

**Figure 3.** β-arrestin 1 (β-arr1), β-arrestin 2 (β-arr2), and Nedd4 are recruited to mGlu7 by agonist stimulation within 5 min. (**A–F**) Time-course interaction among mGlu7, Nedd4, and β-arrestins. HEK 293 T cells were co-transfected with FLAG-mGlu7, HA-Nedd4, and β-arrestin 1-GFP (**A**) or β-arrestin 2-GFP (**B**). In panel E, myc-mGlu7, HA-Nedd4, and β-arrestin 1-FLAG were utilized. L-Glu (1 mM) was administered for the indicated time before cell lysis. Cell lysates were immunoprecipitated with anti-FLAG antibody and western blotting was carried out with the indicated antibodies. Time-course binding of β-arrestins to mGlu7 (**C**), Nedd4 to mGlu7 (**D**), β-arrestin 1 to Nedd4 (**F**) were quantified and presented as mean ± SEM (C, β-arr1, 5 min, 1.79 ± 0.31; 15 min, 1.11 ± 0.23; 30 min, 0.73 ± 0.12; 60 min, 0.52 ± 0.17; β-arr2, 5 min, 1.36 ± 0.10; 15 min, 1.36 ± 0.15; 30 min, 1.01 ± 0.18; 60 min, 0.96 ± 0.38; D, β-arr1, 5 min, 2.36 ± 0.57; 15 min, 1.16 ± 0.31; 30 min, 0.61 ± 0.14; 60 min, 0.72 ± 0.49; β-arr2, 5 min, 2.20 ± 0.36; 15 min, 1.97 ± 0.68; 30 min, 1.96 ± 0.62; 60 min, 1.83 ± 0.49; F, β-arr1, 1 min, 1.67 ± 0.12; 10 min, 1.72 ± 0.19; 30 min, 1.11 ± 0.17; 120 min, 0.71 ± 0.13; n = 3–4, *p<0.05 versus 0 min, Student's *t*-test). (**G**) Endogenous β-arrestin 1, Nedd4, and mGlu7 are present in the same complex. After cortical neurons were treated with 400 μM L-AP4 for 5 min, the lysates were immunoprecipitated with anti-β-arrestin 1 antibody or anti-GFP antibody as a control. Arrow and arrowhead indicate β-arrestin 1 and immunoglobulin heavy chain (IgH), respectively. (**H**) Bar graph represents mean ± SEM showing quantification of mGlu7 or Nedd4 binding levels to β-arrestin 1 (mGlu7 L-AP4, 1.56 ± 0.19; Nedd4 L-AP4, 1.46 ± 0.13; n = 5, *p<0.05 versus vehicle, Student's *t*-test). (**I**) myc-mGlu7 and HA-Nedd4 were co-transfected either with pSuper β-arrestin 1, 2 shRNA or pSuper control shRNA in HEK 293 T cells. Three days after transfection, 1 mM L-Glu was treated for 5 min and cell lysates were immunoprecipitated using anti-myc antibody. Immunoprecipitates were resolved by SDS-PAGE and analyzed by western blotting using the indicated antibodies. (**J**) Bar graph represents mean ± SEM showing Nedd4 binding to mGlu7 levels normalized to untreated Ctl KD (Ctl KD + L-Glu, 1.98 ± 0.22; β-arr1 KD, 1.46 ± 0.23; β-arr1 KD + L-Glu, 1.51 ± 0.39; β-arr2 KD, 1.19 ± 0.13; β-arr2 KD + L-Glu, 0.97 ± 0.25; n = 3–5, *p<0.05, n.s. indicates p>0.05, one-way ANOVA). (**K**) β-arrestins are involved in agonist-induced Nedd4 recruitment to mGlu7 in neurons. Cultured cortical neurons were infected with lentiviruses harboring β-arrestin 1 or 2 shRNA (β-arr 1 or 2 KD), or non-related target shRNA (Ctl KD) for 7 days. Both β-arrestin 1 and 2 shRNA were co-infected for β-arr1/2 KD lane. After treatment with 400 μM L-AP4 for 5 min, the lysates were immunoprecipitated with anti-mGlu7 antibody and endogenous Nedd4 binding was analyzed by western blotting. (**L**) Bar graph represents mean ± SEM showing quantification of Nedd4 binding levels to mGlu7 (Ctl KD + L-AP4, 1.54 ± 0.11; β-arr1 KD + L-AP4, 1.99 ± 0.29; β-arr2 KD + L-AP4, 1.73 ± 0.16; β-arr1/2 KD, 1.15 ± 0.07;

*Figure 3 continued on next page*

*Figure 3 continued*

n = 4, *p<0.05 versus vehicle in Ctl KD, Student's *t*-test). (M) β-arrestins mediate agonist-induced ubiquitination of mGlu7 in neurons. Cultured cortical neurons were infected with β-arrestins shRNA lentiviruses and then incubated with L-AP4 (400 µM) as panel K. Cell lysates were immunoprecipitated with anti-ubiquitin antibody (FK2) and western blotting was performed with anti-mGlu7 antibody. (N) Bar graph represents mean ± SEM showing mGlu7 ubiquitination levels normalized to Ctl KD (β-arr1 KD, 0.77 ± 0.06; β-arr2 KD, 0.74 ± 0.08; β-arr1/2 KD, 0.50 ± 0.03; n = 4, **p<0.01, *p<0.05, Student's *t*-test).

DOI: https://doi.org/10.7554/eLife.44502.007

The following source data and figure supplements are available for figure 3:

**Source data 1.** Binding analysis of mGlu7, Nedd4, and beta-arrestins complex.

DOI: https://doi.org/10.7554/eLife.44502.015

**Figure supplement 1.** Binding of β-arrestins to mGlu7.

DOI: https://doi.org/10.7554/eLife.44502.008

**Figure supplement 2.** The binding affinity of Nedd4 and β-arrestin 1 to mGlu7 is not altered by ubiquitination site mutations of mGlu7.

DOI: https://doi.org/10.7554/eLife.44502.009

**Figure supplement 2—source data 1.** Binding affinity of Nedd4 and beta-arrestin 1 to mGlu7 ubiquitination site mutants.

DOI: https://doi.org/10.7554/eLife.44502.010

**Figure supplement 3.** Co-immunoprecipitation assay to supplement *Figures 3K and L*.

DOI: https://doi.org/10.7554/eLife.44502.011

**Figure supplement 3—source data 1.** Binding of Nedd4 to mGlu7 by beta-arrestins in neurons.

DOI: https://doi.org/10.7554/eLife.44502.012

**Figure supplement 4.** Ubiquitination assay to supplement *Figures 3M and N*.

DOI: https://doi.org/10.7554/eLife.44502.013

**Figure supplement 4—source data 1.** Ubiquitination of mGlu7 by beta-arrestins in neurons.

DOI: https://doi.org/10.7554/eLife.44502.014

(*Flor et al., 1997*). We found that both Nedd4 and β-arrestin 1 bind to the CTs of mGlu7a as well as mGlu7b (*Figure 4A*). Additionally, Nedd4 interacted with the iL2 domain and to a lesser extent with the iL1 domain of mGlu7. While protein phosphatase 1γ1 (PP1γ1) interacted only with mGlu7b CT as previously shown (*Suh et al., 2013*), β-arrestin 1 showed strong interaction with the iL2 domain of mGlu7 (*Figure 4A*). To determine if Nedd4 and β-arrestins directly bind to mGlu7, recombinant His-tagged Nedd4, β-arrestin 1, and β-arrestin 2 proteins were harvested from an *E. coli* expression system and purified using Ni-NTA resin. We performed in vitro binding assays and found that purified Nedd4 protein directly binds to mGlu7 CT and iL2 domains immobilized on Glutathione-Sepharose beads (*Figure 4B*). Both β-arrestin 1 and β-arrestin 2 purified proteins also directly interacted with mGlu7 CTs and iL2 (*Figure 4B*).

We next sought to identify the Nedd4 domain responsible for interaction with mGlu7. As Nedd4 contains 4 WW domains and a HECT domain, we generated FLAG-tagged deletion mutants of Nedd4 in which each domain was deleted by site-directed fragment deletion mutagenesis (*Figure 4C*). We performed co-immunoprecipitation assay with myc-mGlu7, and found that Nedd4 ΔHECT mutant does not interact with mGlu7, indicating the Nedd4 HECT domain is responsible for interaction with mGlu7 (*Figure 4D*).

## Nedd4 and β-arrestins regulate mGlu7 surface expression and agonist-induced endocytosis

We previously reported that PTMs such as phosphorylation and SUMOylation regulate mGlu7 endocytosis in hippocampal neurons (*Choi et al., 2016*; *Suh et al., 2013*; *Suh et al., 2008*). To determine if ubiquitin modification regulates endocytosis of endogenous mGlu7, we utilized MLN7243, a cell-permeable small-molecule inhibitor of ubiquitin-activating enzyme (UAE, also known as E1 enzyme). When we treated cultured cortical neurons with MLN7243, we found surface expression of endogenous mGlu7 is increased while mGlu7 ubiquitination is reduced (*Figure 5—figure supplement 1*). To specifically evaluate the role of Nedd4-mediated ubiquitination on mGlu7 in receptor trafficking, we co-transfected myc-mGlu7 and Nedd4 WT or C867S in cultured hippocampal neurons. We first labeled surface-expressed receptors with anti-myc antibody and allowed endocytosis at 37°C for 15 min in the absence or presence of agonist L-AP4. Remaining receptors on the cell surface were visualized with Alexa Fluor 568-conjugated secondary antibodies before permeabilization (red), and

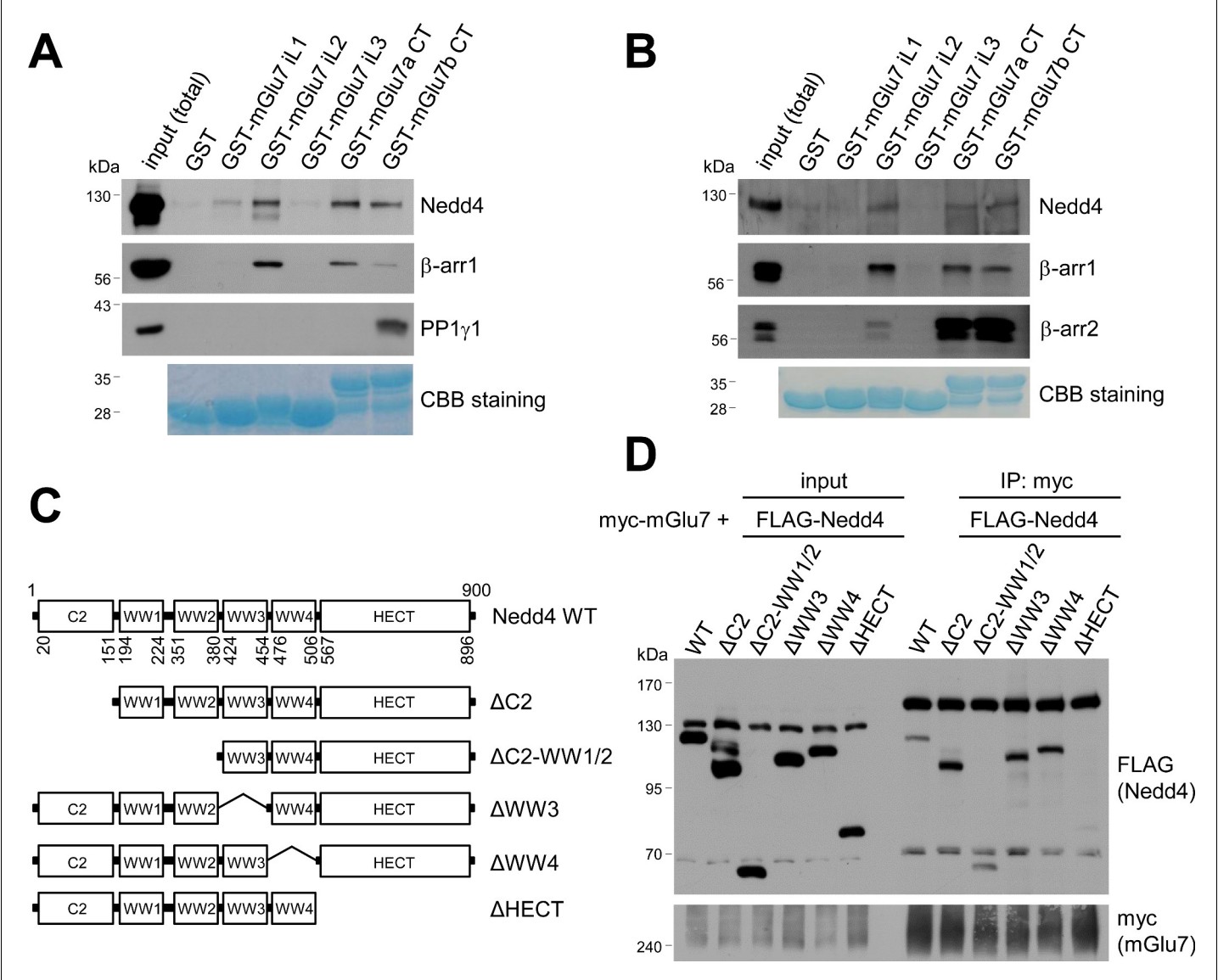

**Figure 4.** Mapping of binding domains between mGlu7 and Nedd4 or β-arrestins. (**A**) GST pull-down experiments to identify the interaction domains of mGlu7 with β-arrestin 1 or Nedd4. Total rat brain extract was subjected to a pull-down assay using immobilized GST fusion proteins containing the mGlu7 iL 1–3 domains, mGlu7a CT, or mGlu7b CT. After washing, bound proteins were analyzed by SDS-PAGE and western blotting using the indicated antibodies. The amount of GST fusion protein as bait was verified by Coomassie Brilliant Blue (CBB) staining. (**B**) Direct binding assay between mGlu7 and Nedd4 or β-arrestins. Recombinant His-tagged Nedd4, β-arrestin 1, or β-arrestin 2 proteins were purified using Ni-NTA resin. The purified proteins were incubated at 4°C for 2 hr with immobilized GST-mGlu7 iL or CT domains. After washing, bound proteins were analyzed by SDS-PAGE and western blotting using the indicated antibodies. (**C**) Mapping of binding domain of Nedd4 to mGlu7. Schematic diagrams of Nedd4 where each domain has been deleted are shown. ΔC2, amino acids (aa) 1–151; ΔC2-WW1/2, aa 1–380; ΔWW3, aa 381–454; ΔWW4, aa 455–506; ΔHECT, aa 507–900; Δ denotes deletion. (**D**) mGlu7 binds to the HECT domain of Nedd4. myc-mGlu7 was co-transfected with the indicated FLAG-Nedd4 domain deletion constructs in HEK 293 T cells. Immunoprecipitation was carried out with anti-myc antibody, and bound proteins were detected with anti-FLAG antibody.

DOI: https://doi.org/10.7554/eLife.44502.016

internalized receptors were labeled with Alexa Fluor 488-conjugated secondary antibodies after permeabilization (green). We observed a marked increase in mGlu7 endocytosis by Nedd4 WT, but not by Nedd4 C867S (*Figure 5A,B*; *Figure 5—figure supplement 2A*). We also found that agonist-induced endocytosis of mGlu7 was impaired by co-expression of Nedd4 shRNA (Nedd4 KD) compared with control shRNA (Ctl KD) in hippocampal neurons (*Figure 5C,D*; *Figure 5—figure*

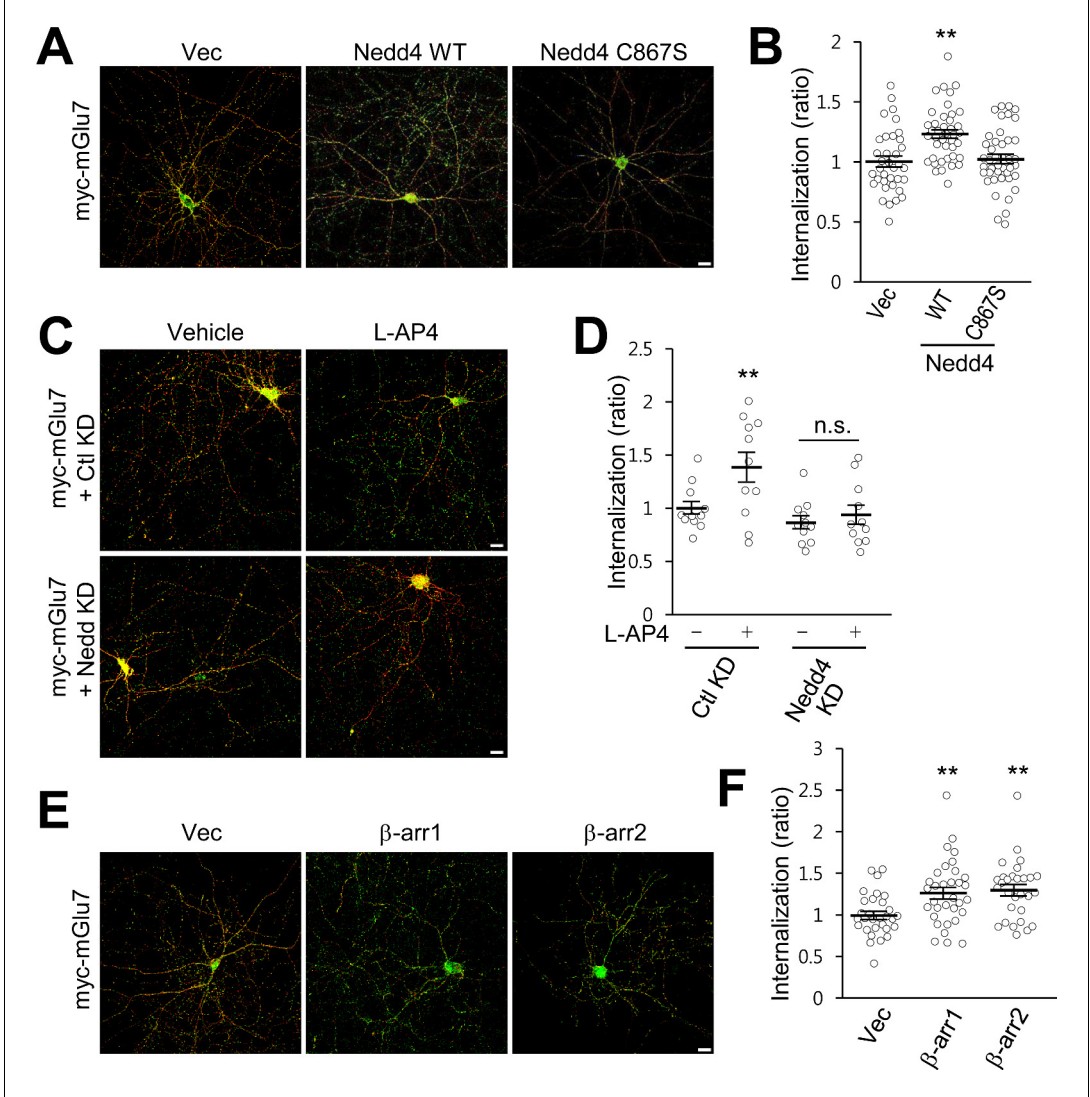

**Figure 5.** β-arrestins and Nedd4 regulate endocytosis of mGlu7 in neurons. (A) The endocytosis of mGlu7 was analyzed by an antibody uptake internalization assay. myc-mGlu7 was co-transfected with Nedd4 WT, C867S mutant, or vector control (Vec) in cultured hippocampal neurons. Two days after transfection, neurons were labeled with anti-myc antibody for 10 min, and returned to conditioned media for 15 min at 37°C. Neurons were fixed and incubated with Alexa Fluor 568-conjugated secondary antibody (red) to label surface-expressed receptors before permeabilization. After permeabilization with 0.25% Trion X-100 for 5 min, neurons were then incubated with Alexa Fluor 488-conjugated secondary antibody (green) to label the internalized receptors. Merged images are presented in which the red signal represents the surface mGlu7 and the green signal represents the internalized mGlu7. Scale bar, 20 μm. (B) Summary histograms quantifying the internalized mGlu7 from panel A are present as the ratio of the internalized population compared with total (surface + internalized) population measured using Metamorph software. Scatter plots show mean ± SEM (Vec, 1.00 ± 0.04; Nedd4 WT, 1.23 ± 0.04; Nedd4 C867S, 1.02 ± 0.04; n > 35, **p<0.01, one-way ANOVA). (C) myc-mGlu7 was co-transfected with pSuper-Ctl shRNA (Ctl KD) or pSuper-Nedd4 shRNA (Nedd4 KD) in cultured hippocampal neurons. Internalization of mGlu7 was analyzed in the absence or presence of 400 μM L-AP4 for 15 min at 37°C. (D) Summary histograms quantifying the internalized mGlu7 from panel C. Scatter plots show mean ± SEM (Vec, 1.00 ± 0.06; Vec + L-AP4, 1.39 ± 0.14; Nedd4 KD, 0.86 ± 0.06; Nedd4 KD + L-AP4, 0.94 ± 0.09; n > 10, *p<0.05, n.s. indicates p>0.05, one-way ANOVA). (E) myc-mGlu7 and β-arrestin 1 or 2 were co-expressed and the internalized mGlu7 was analyzed as above. (F) Summary histograms quantifying the internalized mGlu7 from panel E. Scatter plots show mean ± SEM (Vec, 1.00 ± 0.05; β-arr1, 1.26 ± 0.07; β-arr2, 1.30 ± 0.07; n > 30, **p<0.01, one-way ANOVA).

DOI: https://doi.org/10.7554/eLife.44502.017

The following source data and figure supplements are available for figure 5:

**Source data 1.** Endocytosis of mGlu7 by beta-arrestins and Nedd4.
DOI: https://doi.org/10.7554/eLife.44502.020

**Figure supplement 1.** Ubiquitination is required for endocytosis of endogenous mGlu7.

*Figure 5 continued on next page*

*Figure 5 continued*

DOI: https://doi.org/10.7554/eLife.44502.018

**Figure supplement 2.** Separated images that supplement the merged images shown in *Figure 5*.

DOI: https://doi.org/10.7554/eLife.44502.019

*supplement 2B*). These results indicate that agonist-induced endocytosis of mGlu7 is regulated by Nedd4-mediated ubiquitin modification of receptor. It has been known that β-arrestins regulate GPCR endocytosis by forming complexes with clathrin-coated structures following agonist-induced activation of receptors (*Eichel et al., 2016*; *Jean-Charles et al., 2016*). Thus, we examined if β-arrestins also could regulate the endocytosis of mGlu7. When β-arrestin 1 or 2 was co-expressed with mGlu7, we found a profound increase in internalization of mGlu7 by either β-arrestin 1 or 2 in cultured hippocampal neurons (*Figure 5E,F*; *Figure 5—figure supplement 2C*). These results indicate agonist-induced mGlu7 ubiquitination by the Nedd4-β-arrestin complex promotes receptor endocytosis in hippocampal neurons.

## β-arrestin 2 and Nedd4 regulate MAPK signaling of mGlu7 in neurons

It has been proposed that β-arrestins initiate β-arrestin-dependent signaling to ERK1/2, one of the MAPKs in the plasma membrane and endosomes (*Eichel et al., 2018*; *Nuber et al., 2016*; *Thomsen et al., 2016*). Although L-AP4-induced ERK activation has been reported to require β-arrestin 2 but not β-arrestin 1 in neurons (*Gu et al., 2012*; *Jiang et al., 2006*), Nedd4-mediated ERK signaling remains unknown. To test whether β-arrestins or Nedd4 are required for ERK activation in response to agonist stimulation of mGlu7, we knocked down β-arrestin 1, 2, or Nedd4 via shRNA lentivirus in cultured hippocampal neurons and treated them with L-AP4. We observed that ERK1/2 phosphorylation significantly increased after 5 min treatment with L-AP4 in Ctl KD or β-arr1 KD neurons, whereas β-arr2 KD or Nedd4 KD did not induce agonist-induced ERK1/2 phosphorylation (*Figure 6A,B*), indicating the necessity for β-arrestin 2 and Nedd4 in mGlu7-mediated ERK signaling. We observed a similar impairment of agonist-induced c-Jun N-terminal kinases (JNKs) activation in β-arr2 KD neurons. Either β-arr1 KD or Nedd4 KD showed little effect on mGlu7-mediated JNK signaling (*Figure 6A,C*). These results suggest that mGlu7-dependent ERK signaling is regulated by β-arrestin 2 and depends on Nedd4-mediated ubiquitination, whereas JNK signaling is independent of Nedd4-mediated ubiquitination in neurons.

## Nedd4-mediated ubiquitination leads to degradation of mGlu7 by both the lysosomal and proteasomal degradation pathways

During the experiments to analyze mGlu7 ubiquitination, we observed a marked decrease in total expression of mGlu7 when agonist L-Glu was administered for more than 30 min or Nedd4 was co-transfected (*Figure 1B,E*). As is evident in *Figure 7A*, total expression of mGlu7 was reduced when Nedd4 was co-expressed with mGlu7. However, co-expression of Nedd4 C867S had little effect on total expression of mGlu7 (*Figure 7A*). The total expression level of mGlu7 was reduced by L-Glu treatment for 60 min, which was blocked by co-expression of Nedd4 C867S (*Figure 7B*). These results indicate that the surface stability of mGlu7 is determined by Nedd4-mediated ubiquitin modification of mGlu7. To further investigate the degradation pathway of ubiquitinated mGlu7, we treated mGlu7-transfected HEK 293 T cells with MG132 (proteasome inhibitor) or leupeptin (lysosome inhibitor) for 2 hr prior to treatment with L-Glu. We found that agonist-induced mGlu7 degradation was significantly blocked by leupeptin treatment (*Figure 7C,D*). This result indicates that ubiquitination regulates the intracellular sorting of mGlu7 into early and late endosomes, and subsequent degradation in lysosomes. Furthermore, we were able to observe that MG132 also inhibits agonist-induced mGlu7 degradation (*Figure 7C,D*), implying that ubiquitin-proteasome system is also involved in the control of mGlu7 protein homeostasis. To determine the involvement of Nedd4 in mGlu7 stability in neurons, Nedd4 WT or Nedd4 C867S was expressed on cultured cortical neurons via lentivirus. We found that Nedd4 WT significantly reduced mGlu7 expression in neurons, whereas Nedd4 C867S did not. In addition, Nedd4 WT-mediated mGlu7 degradation was blocked by MG132 or leupeptin (*Figure 7E,F*). Thus, Nedd4-mediated ubiquitination is required for mGlu7

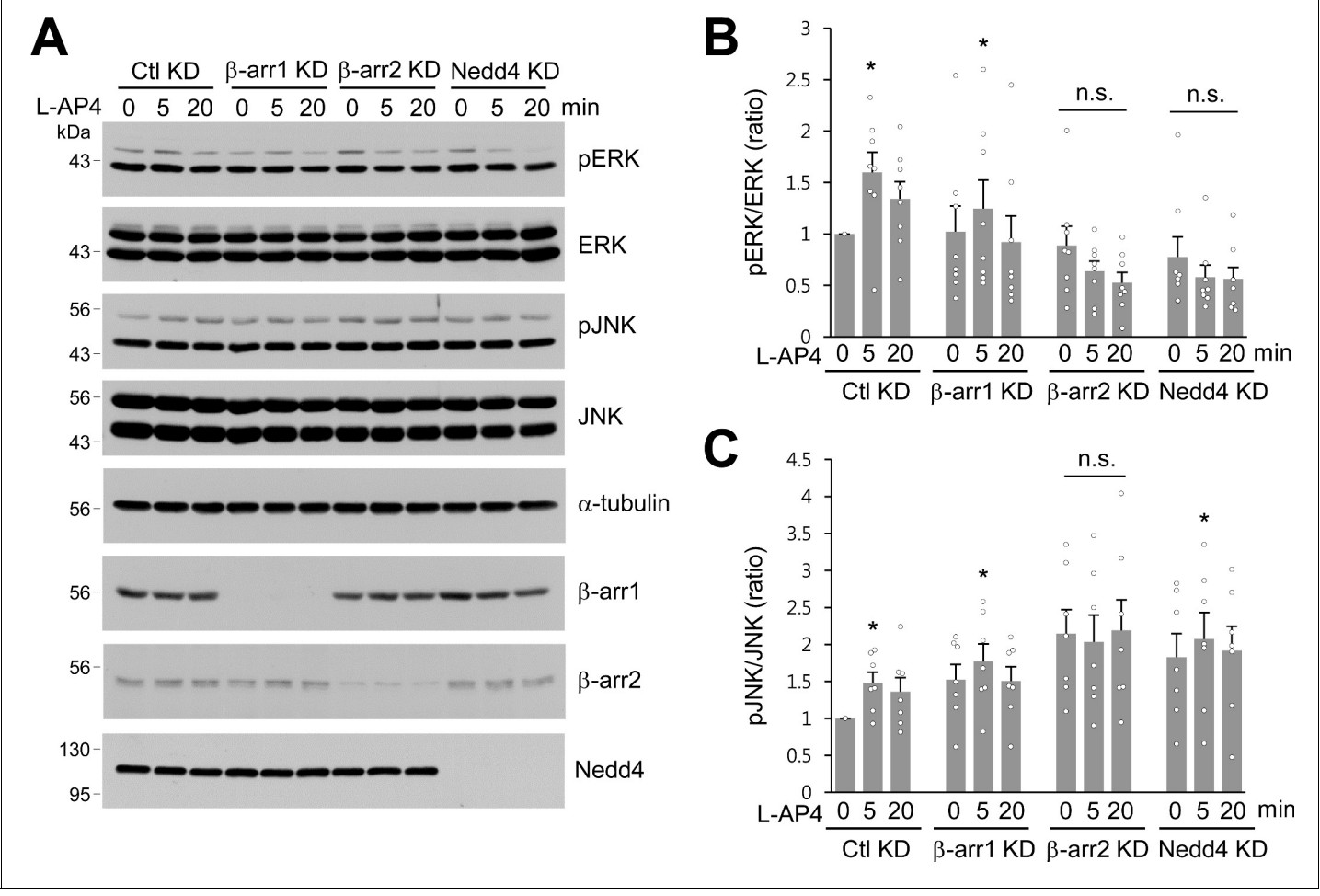

**Figure 6.** β-arrestins and Nedd4 regulate ERK and JNK signaling of mGlu7 in neurons. (**A**) Cultured hippocampal neurons were infected with control (Ctl KD), β-arrestin 1 (β-arr1 KD), β-arrestin 2 (β-arr2 KD), or Nedd4 KD shRNA lentiviruses for 7 days. At DIV 14, neurons were treated with 400 µM L-AP4 for 0, 5, 20 min, and neuronal lysates were separated by SDS-PAGE and probed using the indicated antibodies. (**B**) Bar graph represents mean ± SEM of pERK band intensities normalized to the control lane (Ctl KD, 5 min, 1.60 ± 0.20; 20 min, 1.34 ± 0.17; β-arr1 KD, 0 min, 1.02 ± 0.25; 5 min, 1.25 ± 0.28; 20 min, 0.92 ± 0.26; β-arr2 KD, 0 min, 0.89 ± 0.19; 5 min, 0.64 ± 0.10; 20 min, 0.53 ± 0.10; Nedd4 KD, 0 min, 0.78 ± 0.20; 5 min, 0.58 ± 0.12; 20 min, 0.56 ± 0.11; n = 8, *p<0.05, n.s. indicates p>0.05, Student's *t*-test). (**C**) Bar graph represents mean ± SEM of pJNK band intensities normalized to the control lane (Ctl KD 5 min, 1.48 ± 0.14; 20 min, 1.36 ± 0.19; β-arr1 KD 0 min, 1.52 ± 0.21; 5 min, 1.77 ± 0.24; 20 min, 1.50 ± 0.19; β-arr2 KD 0 min, 2.15 ± 0.33; 5 min, 2.03 ± 0.36; 20 min, 2.19 ± 0.41; Nedd4 KD 0 min, 1.83 ± 0.32; 5 min, 2.07 ± 0.36; 20 min, 1.92 ± 0.33; n = 7, *p<0.05, n.s. indicates p>0.05, Student's *t*-test).

DOI: https://doi.org/10.7554/eLife.44502.021

The following source data is available for figure 6:

**Source data 1.** ERK and JNK signaling of mGlu7 by beta-arrestins and Nedd4.

DOI: https://doi.org/10.7554/eLife.44502.022

degradation, which is accomplished through the proteasomal and lysosomal degradation pathways in neurons.

## Discussion

Ubiquitination has been originally considered as a signal that directs non-lysosomal protein degradation to the 26S proteasome, however it has emerged as a critical PTM with multiple roles that directly or indirectly govern trafficking, signaling, and lysosomal degradation of GPCRs (**Dores and Trejo, 2012**; **Mukhopadhyay and Riezman, 2007**). After agonist-induced activation of GPCRs triggers the coupling of heterotrimeric G proteins and subsequent second messenger signaling,

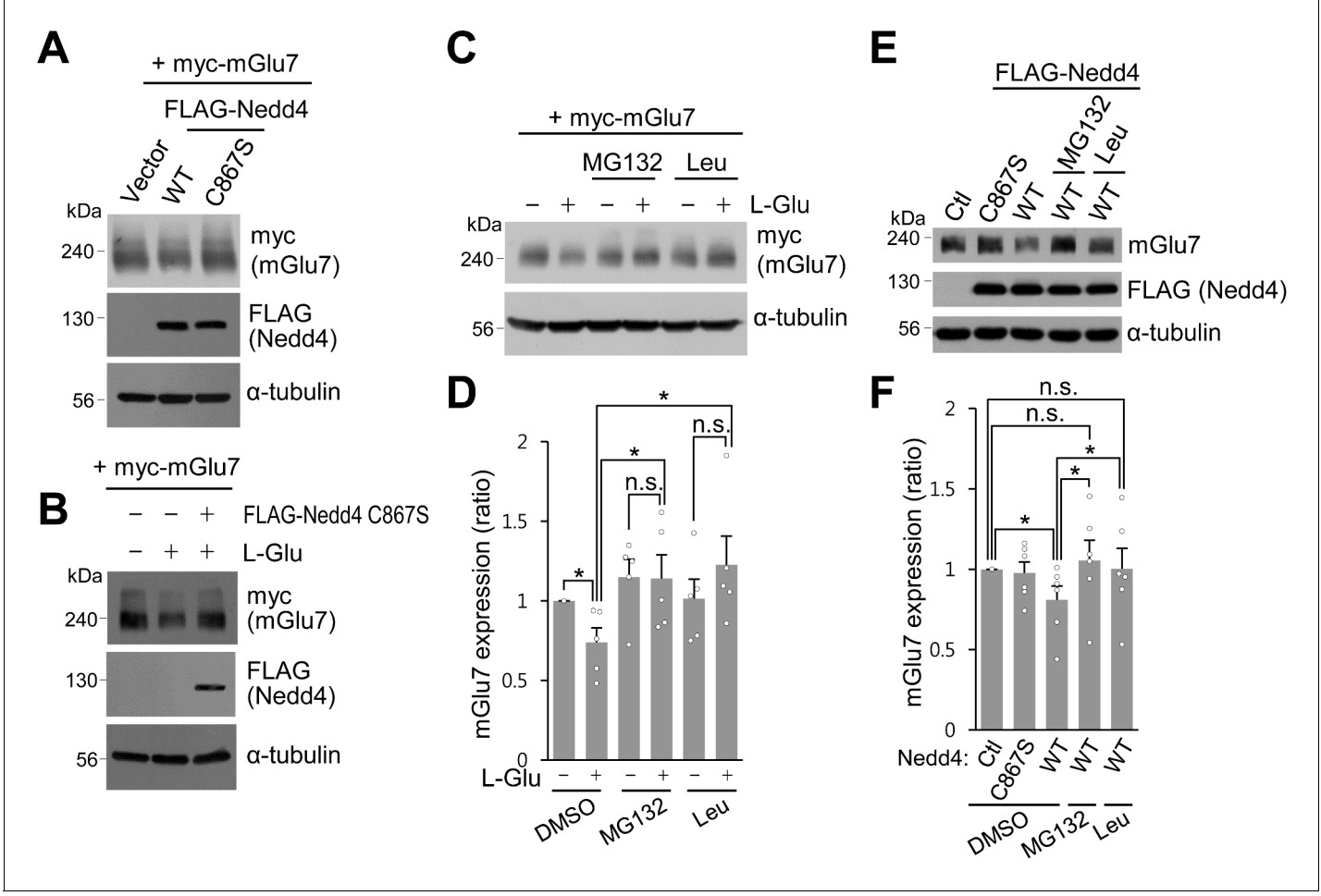

**Figure 7.** Nedd4-mediated or agonist-stimulated mGlu7 degradation occurs via both the proteasomal and lysosomal degradation pathways. (**A**) myc-mGlu7 and FLAG-Nedd4 WT or C867S were co-transfected in HEK 293 T cells. Thirty-six hours after transfection, cell lysates were analyzed by western blotting using the indicated antibodies. (**B**) Following transfection, cells were starved overnight in serum-free DMEM culture medium and then treated with 1 mM L-Glu for 1 hr. Cell lysates were analyzed by western blotting using the indicated antibodies. (**C**) HEK 293 T cells transiently expressing myc-tagged mGlu7 was incubated with 100 μM MG132 or 52.5 μM leupeptin (Leu) for 2 hr and then L-Glu was added for 1 hr. Cell lysates were analyzed by SDS-PAGE and western blotting using the indicated antibodies. (**D**) Summary histograms quantifying mGlu7 expression in panel C. Bar graph represents mean ± SEM of band intensities normalized to control lane (DMSO + L-Glu, 0.74 ± 0.09; MG132, 1.15 ± 0.11; MG132 + L-Glu, 1.14 ± 0.15; Leu, 1.01 ± 0.12; Leu + L-Glu, 1.23 ± 0.18; n = 5, *p<0.05, n.s. indicates p>0.05, Student's *t*-test). (**E**) FLAG-tagged Nedd4 WT or Nedd4 C867S was expressed on cultured cortical neurons via lentiviruses. After treatment with 0.5 μg/ml cycloheximide for 21 hr, the neurons were incubated with 100 μM MG132 or 52.5 μM Leu for 4 hr and then 400 μM L-AP4 was added for 1 hr. Cell lysates were analyzed by western blotting using the indicated antibodies. (**F**) Summary histograms quantifying mGlu7 expression in panel E. Bar graph represents mean ± SEM of band intensities normalized to control lane (Nedd4 C867S, 0.98 ± 0.07; Nedd4 WT, 0.81 ± 0.09; Nedd4 WT + MG132, 1.06 ± 0.13; Nedd4 WT + Leu, 1.00 ± 0.13; n = 5, *p<0.05, n.s. indicates p>0.05, Student's *t*-test).

DOI: https://doi.org/10.7554/eLife.44502.023

The following source data is available for figure 7:

**Source data 1.** Degradation pathways of ubiquitinated mGlu7.
DOI: https://doi.org/10.7554/eLife.44502.024

recruitment of β-arrestins to GPCRs plays a classical role in receptor desensitization. Recent studies have revealed that β-arrestins modulate multifaceted functions in endocytosis, signaling, and ubiquitination of GPCRs by scaffolding the clathrin-mediated endocytosis machinery, MAPKs, or E3 ubiquitin ligases such as Nedd4, MDM2, and AIP4 (*Jean-Charles et al., 2016*; *Ranjan et al., 2017*; *Shenoy and Lefkowitz, 2011*; *Smith and Rajagopal, 2016*; *Srivastava et al., 2015*). In the present study, we have elucidated several key molecular features regulating mGlu7 ubiquitination by forming

a complex with β-arrestins and Nedd4 E3 ligase in an activity-dependent manner. We found that mGlu7 undergoes constitutive and agonist-induced ubiquitination in the lysine residues of both iL2 and CT, which is primarily mediated by Nedd4 E3 ligase. Upon agonist stimulation, β-arrestins are able to recruit Nedd4 to mGlu7 and facilitate ubiquitination of mGlu7. Nedd4 and β-arrestins regulate constitutive and agonist-induced endocytosis of mGlu7, and are required for MAPK signaling of mGlu7 in neurons. Finally, Nedd4-mediated mGlu7 ubiquitination regulates receptor stability through both the lysosomal and the proteasomal degradation pathways.

The E3 ubiquitin ligase is a key determinant of substrate specificity by catalyzing the transfer of ubiquitin from E2 enzymes to target proteins. Little is known about E3 ligases about their role in regulating the ubiquitination of mGlu receptors. Siah-1A (seven in absentia homolog-1A) is the first identified E3 ubiquitin ligase for Group 1 mGlu receptors (*Moriyoshi et al., 2004*). Siah-1A directly binds to Group 1 mGlu receptors and mediates receptor ubiquitination, resulting in receptor degradation via the ubiquitin-proteasome pathway. Siah-1A also regulates agonist-induced endocytosis of mGlu5 by displacing calmodulin binding, and promotes the lysosomal degradation of mGlu5 (*Gulia et al., 2017*; *Ishikawa et al., 1999*; *Ko et al., 2012*). In contrast, we have identified Nedd4 as a primary E3 ligase for ubiquitination of mGlu7, a presynaptically expressed Group III mGlu receptor. Although mGlu7 does not possess the PPxY motif, a canonical Nedd4 binding motif, Nedd4 directly binds to the iL2 and CT domains of mGlu7 by the HECT domain of Nedd4. In addition to modulating mGlu7-mediated ERK signaling, Nedd4 regulates mGlu7 trafficking by inducing ubiquitination, resulting in endocytosis and degradation of surface-expressed mGlu7 into proteasomes and lysosomes. We observed a marked induction of mGlu7 ubiquitination when exogenous ubiquitin was co-expressed with mGlu7 in HEK 293 T cells. In contrast, endogenous mGlu7 ubiquitination induced by agonist stimulation was less prominent in neurons (*Figure 1C,F*). This result likely occurs because only the surface-expressed fraction of mGlu7 can be ubiquitinated directly from agonist stimulation. We have observed that agonist stimulation increases ubiquitination of the surface-expressed mGlu7 to some degree but not ubiquitination of the intracellular mGlu7 (*Figure 1D*). We were not able to observe more prominent endogenous mGlu7 ubiquitination with longer agonist incubations for up to 2 hr or by other neuronal stimulation such as bicuculline or KCl (data not shown). Of note, we observed some remaining levels of mGlu7 ubiquitination despite the loss of Nedd4 in *Figure 1F*. This result suggests that other E3 ligases than Nedd4 may play a role in mGlu7 ubiquitination in neurons.

We identified two binding sites, iL2 and CT in mGlu7 that interact with β-arrestins and Nedd4. Recently, a crystal structure of rhodopsin-arrestin complex revealed that a conserved serine/threonine-rich domain, Px(x)PxxP/E/D (P, phospho-serine or phospho-threonine; X, any amino acid) across the major GPCR subfamilies serves as a common phosphorylation code for arrestin recruitment (*Zhou et al., 2017*). A serine/threonine rich domain on mGlu7 CT ($_{868}$-TAATMSSRLS-$_{877}$) is compatible with this phosphorylation code for arrestin binding, and is therefore a candidate β-arrestin-interacting domain. In addition, the iL2 domain of GPCRs has been shown to contribute to association with β-arrestin by providing a partial core contact site for both Gα and β-arrestin (*Hilger et al., 2018*; *Latorraca et al., 2018*; *Ranjan et al., 2017*). It was shown that the iL2 domain of mGlu receptors including mGlu8 determines G-protein coupling selectivity (*Francesconi and Duvoisin, 1998*; *Gomeza et al., 1996*; *Havlickova et al., 2003*; *Pin et al., 1994*). Furthermore, in addition to the CT domain, the iL2 domain of mGlu1 is a site for GRK2 binding and GRK2-mediated signaling (*Dhami et al., 2005*). Accordingly, additional binding of β-arrestin-Nedd4 complex to mGlu7 iL2 may sterically hinder G-protein binding and provide a platform for a receptor core for fully-engaged binding of β-arrestin to mGlu7. Although we observed β-arrestins and Nedd4 bind constitutively to mGlu7, binding affinity of Nedd4 or β-arrestin 1 to mGlu7 is not affected by ubiquitination site mutations in neurons (*Figure 3—figure supplement 2*). This result suggests that mGlu7 ubiquitination per se is not a prerequisite for the complex formation.

It has been expected that β-arrestin 1 and β-arrestin 2 are functionally redundant due to the high degree of sequence and structural similarity, however recent studies have suggested their partially distinct functional roles (*Srivastava et al., 2015*). For example, the desensitization, ubiquitination, and endocytosis of β2AR is primarily mediated by β-arrestin 2, not by β-arrestin 1 (*Han et al., 2013*; *Jean-Charles et al., 2016*; *Shenoy and Lefkowitz, 2011*). AT$_{1a}$R-mediated ERK signaling is eliminated by β-arrestin 2 KD in HEK 293 cells, while β-arrestin 1 KD augments ERK signaling (*Ahn et al., 2004*; *Srivastava et al., 2015*). Several studies have reported the controversial effects of β-arrestins

in the endocytosis of Group I mGlu receptors (*Iacovelli et al., 2013*; *Suh et al., 2018*). Constitutive endocytosis of mGlu1 was β-arrestin 1-independent (*Dale et al., 2001*) or β-arrestin 1-dependent (*Pula et al., 2004*) in the heterologous cell populations. Agonist-induced mGlu1 endocytosis was β-arrestin 1- and 2-dependent (*Mundell et al., 2001*), or β-arrestin 1-dependent only under co-expression with GRKs in heterologous cells (*Dale et al., 2001*). In contrast, β-arrestin 1 does not seem to directly participate in agonist-induced mGlu1 endocytosis (*Iacovelli et al., 2003*). Our current study showed that β-arrestin 1 and 2 exhibit functional redundancy for mGlu7, while β-arrestin 2 seems to have a lower binding affinity to mGlu7 than β-arrestin 1. Both β-arrestin 1 and 2 interact constitutively with mGlu7 and are recruited to mGlu7 by agonist stimulation. Although both β-arrestin 1 and 2 KD were required to diminish agonist-induced Nedd4 recruitment to mGlu7 in neurons, agonist-induced mGlu7 ubiquitination was reduced by either the loss of β-arrestin 1 or 2 (*Figure 3K–N*). It is unclear why the requirement for β-arrestins is different between Nedd4 recruitment to mGlu7 and agonist-induced mGlu7 ubiquitination. However, we postulate that β-arrestins may recruit isoform-specific ubiquitin E3 ligases other than Nedd4 in neurons.

Both β-arrestin 1 and 2 are required for constitutive endocytosis of mGlu7 in cultured hippocampal neurons. In addition, our study has demonstrated that β-arrestin 2 rather than β-arrestin 1 is involved in mGlu7-mediated ERK and JNK signaling in hippocampal neurons (*Figure 6*). In particular, Nedd4-mediated ubiquitination may be involved in ERK signaling rather than JNK signaling. In contrast, Iacovelli et al. has reported the opposite effect of β-arrestin 1 versus β-arrestin 2 on mGlu7-mediated signaling in HEK 293 cells. They showed that β-arrestin 1 increases or decreases agonist-induced ERK or JNK signaling, respectively, while β-arrestin 2 exerts the opposite effects (*Iacovelli et al., 2014*). This discrepancy may be caused by the difference in the expression system of primary neurons versus heterologous cells.

Ubiquitin modification is involved in the endocytosis, recycling, and degradation of transmembrane proteins. Upon ubiquitination, ligand-stimulated receptors on the plasma membrane are primarily degraded by the lysosome, whereas most of the ubiquitinated proteins in the cytoplasm are degraded by the proteasome (*Foot et al., 2017*). Our present study showed that ligand-induced surface mGlu7 degradation is regulated by the ubiquitin-proteasome pathway as well as the lysosomal degradation pathway (*Figure 7*). It may be possible that ubiquitinated mGlu7 can be directly translocated to the proteasome or ubiquitinated β-arrestin-Nedd4 can guide mGlu7 as a complex to the proteasome for degradation. Similar to our results, it has been reported that proteasomal activity is required for ligand-induced degradation of the interleukin-2 receptor, platelet-derived growth factor β receptor, Met tyrosine kinase receptor, epidermal growth factor receptor, growth hormone receptor, tropomyosin-regulated kinase A (TrkA) receptor, and AMPA receptor (*Geetha and Wooten, 2008*; *Jeffers et al., 1997*; *Kesarwala et al., 2009*; *Lin et al., 2011*; *Lin and Man, 2013*; *Mori et al., 1995*; *van Kerkhof et al., 2000*; *Yu and Malek, 2001*).

In conclusion, β-arrestins and Nedd4 play pivotal roles in regulating mGlu7 ubiquitination, endocytosis, signaling, and stability in heterologous cells and neurons. These findings will offer novel mechanistic insights for developing selective drugs with therapeutic potentials for mGlu7-related neuropsychiatric disorders.

## Materials and methods

**Key resources table**

| Reagent type (species) or resource | Designation | Source or reference | Identifiers | Additional information |
|---|---|---|---|---|
| Recombinant DNA reagent | pRK5-myc-mGlu7 WT | Dr. Katherine W. Roche (NIH) | | |
| Recombinant DNA reagent | pRK5-FLAG-mGlu7 WT | This study | | |
| Recombinant DNA reagent | pRK5-myc-mGlu7 Δ857 | This study | | |
| Recombinant DNA reagent | pRK5-myc-mGlu7 Δ860 | This study | | |

*Continued on next page*

*Continued*

| Reagent type (species) or resource | Designation | Source or reference | Identifiers | Additional information |
|---|---|---|---|---|
| Recombinant DNA reagent | pRK5-myc-mGlu7 Δ879 | This study | | |
| Recombinant DNA reagent | pRK5-myc-mGlu7 Δ893 | This study | | |
| Recombinant DNA reagent | pRK5-myc-mGlu7 CT 8K8R | This study | | |
| Recombinant DNA reagent | pRK5-myc-mGlu7 iL 4K4R | This study | | |
| Recombinant DNA reagent | pRK5-myc-mGlu7 12K12R | This study | | |
| Recombinant DNA reagent | pRK5-myc-mGlu7 iL K688/689R | This study | | |
| Recombinant DNA reagent | pRK5-myc-mGlu7 Δ857/iL 4K4R | This study | | |
| Recombinant DNA reagent | pRK5-myc-mGlu7 Δ857/IL K688/689R | This study | | |
| Recombinant DNA reagent | FLAG-ß2AR | Addgene | RRID:Addgene_14697 | |
| Recombinant DNA reagent | pRK5-HA-Ub WT | Addgene | RRID:Addgene_17608 | |
| Recombinant DNA reagent | pRK5-HA-Ub K48 | Addgene | RRID:Addgene_17605 | |
| Recombinant DNA reagent | pRK5-HA-Ub K63 | Addgene | RRID:Addgene_17606 | |
| Recombinant DNA reagent | pGFP-N3-ß-arrestin1-GFP | Dr. Katherine W. Roche (NIH) | | |
| Recombinant DNA reagent | pGFP-N3-ß-arrestin2-GFP | Dr. Katherine W. Roche (NIH) | | |
| Recombinant DNA reagent | pcDNA3-ß-arrestin1-FLAG | Addgene | RRID:Addgene_14687 | |
| Recombinant DNA reagent | pCI-HA-Nedd4 WT | Addgene | RRID:Addgene_27002 | |
| Recombinant DNA reagent | pCI-HA-Nedd4 C867S | This study | | |
| Recombinant DNA reagent | FLAG-Nedd4 WT | This study | | |
| Recombinant DNA reagent | FLAG-Nedd4 C867S | This study | | |
| Recombinant DNA reagent | FLAG-Nedd4 ΔC2 | This study | | |
| Recombinant DNA reagent | FLAG-Nedd4 ΔC2-WW1/2 | This study | | |
| Recombinant DNA reagent | FLAG-Nedd4 ΔWW3 | This study | | |
| Recombinant DNA reagent | FLAG-Nedd4 ΔWW4 | This study | | |
| Recombinant DNA reagent | FLAG-Nedd4 ΔHECT | This study | | |
| Recombinant DNA reagent | pSuper-rat Nedd4 KD | This study | | |
| Recombinant DNA reagent | pSuper-human ß-arrestin1 KD | This study | | |

*Continued on next page*

*Continued*

| Reagent type (species) or resource | Designation | Source or reference | Identifiers | Additional information |
|---|---|---|---|---|
| Recombinant DNA reagent | pSuper-human ß-arrestin2 KD | This study | | |
| Recombinant DNA reagent | pSuper-rat ß-arrestin1 KD | This study | | |
| Recombinant DNA reagent | pSuper-rat ß-arrestin2 KD | This study | | |
| Recombinant DNA reagent | pSuper-non-related target KD | This study | | |
| Recombinant DNA reagent | FHUGW-rat ß-arrestin1 KD | This study | | |
| Recombinant DNA reagent | FHUGW-rat ß-arrestin2 KD | This study | | |
| Recombinant DNA reagent | FHUGW-non-related target KD | This study | | |
| Recombinant DNA reagent | FHUGW-myc-mGlu7 WT | This study | | |
| Recombinant DNA reagent | FHUGW-myc-mGlu7 CT 8K8R | This study | | |
| Recombinant DNA reagent | FHUGW-myc-mGlu7 iL 4K4R | This study | | |
| Recombinant DNA reagent | FHUGW-myc-mGlu7 12K12R | This study | | |
| Recombinant DNA reagent | FHUGW-FLAG-Nedd4 WT | This study | | |
| Recombinant DNA reagent | FHUGW-FLAG-Nedd4 C867S | This study | | |
| Recombinant DNA reagent | pGEX-4T-1-mGlu7 iL1 | This study | | |
| Recombinant DNA reagent | pGEX-4T-1-mGlu7 iL2 | This study | | |
| Recombinant DNA reagent | pGEX-4T-1-mGlu7 iL3 | This study | | |
| Recombinant DNA reagent | pGEX-4T-1-mGlu7a CT | This study | | |
| Recombinant DNA reagent | pGEX-4T-1-mGlu7b CT | This study | | |
| Recombinant DNA reagent | pET28a-Nedd4 | This study | | |
| Recombinant DNA reagent | pET28b-Barr1 | This study | | |
| Recombinant DNA reagent | pET28b-Barr2 | This study | | |
| Antibody | HA (clone 16B12) | BioLegend | cat# 901501 RRID:AB_2565006 | 1:1000, western blotting (WB) |
| Antibody | c-myc (clone 9E10) | SIGMA | cat# M5546 RRID:AB_260581 | 1:1000, WB; 1:500, immunofluorescence staining (IF) |
| Antibody | FLAG (clone M2) | SIGMA | cat# F1804 RRID:AB_262044 | 1:1000, WB |
| Antibody | FLAG | SIGMA | cat# F7425 RRID:AB_439687 | 1:1000, WB |

*Continued on next page*

*Continued*

| Reagent type (species) or resource | Designation | Source or reference | Identifiers | Additional information |
|---|---|---|---|---|
| Antibody | mGlu7a | EMD Millipore | cat# 07–239 RRID:AB_310459 | 1:2000, WB |
| Antibody | Ub (clone FK2) | EMD Millipore | cat# 04–263 RRID:AB_612093 | 1:1000, WB |
| Antibody | Ub (clone P4D1) | Santa Cruz Biotechnology | cat# sc-8017 RRID:AB_2762364 | 1:1000, WB |
| Antibody | Nedd4 | EMD Millipore | cat# 07–049 RRID:AB_310351 | 1:10000, WB |
| Antibody | Nedd4 | R&D SYSTEMS | cat# MAB6218 RRID:AB_10920762 | 1:1000, WB |
| Antibody | Beta-arrestin 1 (clone E274) | Abcam | cat# ab32099 RRID:AB_722896 | 1:1000, WB |
| Antibody | Beta-arrestin 2 (clone C16D9) | Cell Signaling Technology | cat# 3857 RRID:AB_2258681 | 1:1000, WB |
| Antibody | GFP (clone B2) | Santa Cruz Biotechnology | cat# sc-9996 RRID:AB_627695 | 1:500, WB |
| Antibody | GFP | Thermo Fischer Scientific | cat# A11122 RRID:AB_221569 | 1:2000, WB |
| Antibody | α-tubulin | SIGMA | cat# T6199 RRID:AB_477583 | 1:5000, WB |
| Antibody | PP1γ1 | EMD Millipore | cat# 07–1218 RRID:AB_1977432 | 1:500, WB |
| Antibody | phospho-ERK | Santa Cruz Biotechnology | cat# sc-7383 RRID:AB_627545 | 1:1000, WB |
| Antibody | ERK | Santa Cruz Biotechnology | cat# sc-94 RRID:AB_2140110 | 1:2000, WB |
| Antibody | phospho-JNK | Cell Signaling Technology | cat# 4671 RRID:AB_331338 | 1:1000, WB |
| Antibody | JNK | Cell Signaling Technology | cat# 9252 RRID:AB_2250373 | 1:1000, WB |
| Antibody | Goat anti-Mouse IgG (H+L) antibody, Alexa Fluor 568 | Thermo Fischer Scientific | cat# A11031 RRID:AB_144696 | 1:500, IF |
| Antibody | Goat anti-Mouse IgG (H+L) antibody, Alexa Fluor 488 | Thermo Fischer Scientific | cat# A11029 RRID:AB_138404 | 1:500, IF |
| Chemical compound | Normal goat serum blocking solution | VECTOR LABORATORIES | cat# S-1000 RRID:AB_2336615 | |
| Chemical compound | Paraformaldehyde | SIGMA | cat# 158127 | |
| Chemical compound | ProLong Antifade Kit | Thermo Fischer Scientific | cat# P7481 | |
| Chemical compound | L-Glutamic acid | TOCRIS | cat# 0218 | |
| Chemical compound | L-AP4 | TOCRIS | cat# 0103 | |
| Chemical compound | N-Ethylmaleimide (NEM) | SIGMA | cat# 04259 | |
| Chemical compound | MG132 | TOCRIS | cat# 1748 | |
| Chemical compound | Leupeptin hemisulfate | TOCRIS | cat# 1167 | |
| Chemical compound | Cycloheximide | SIGMA | cat# C7698 | |

*Continued on next page*

*Continued*

| Reagent type (species) or resource | Designation | Source or reference | Identifiers | Additional information |
|---|---|---|---|---|
| Chemical compound | MLN7243 | CHEMIETEK | cat# CT-M7243 | |
| Chemical compound | Protein G Sepharose 4 Fast Flow | GE Healthcare | cat# 17061801 | |
| Chemical compound | Protein A Sepharose, from *Staphylococcus aureus* | SIGMA | cat# P3391 | |
| Chemical compound | Glutathione Sepharose 4B GST-tagged resin | GE Healthcare | cat# 17075601 | |
| Chemical compound | EZ-Link Sulfo-NHS-SS-Biotin | Thermo Fischer Scientific | cat# 21331 | |
| Chemical compound | Pierce Streptavidin Agarose | Thermo Fischer Scientific | cat# 20347 | |
| Chemical compound | shRNA to rat Nedd4 | *Schwarz et al., 2010* | | Oligonucleotides |
| Chemical compound | shRNA to rat beta-arrestin 1 | *Simard et al., 2013* | | Oligonucleotides |
| Chemical compound | shRNA to rat beta-arrestin 2 | *Molteni et al., 2009* | | Oligonucleotides |
| Chemical compound | shRNA to human beta-arrestin 1 | *Shenoy et al., 2008* | | Oligonucleotides |
| Chemical compound | shRNA to human beta-arrestin 2 | *Skånland et al., 2009* | | Oligonucleotides |
| Chemical compound | shRNA to non-related target | GenScript | | Oligonucleotides |
| Cell line (human) | HEK293T | ATCC | cat# CRL-3216 RRID:CVCL_0063 | |
| Rat (*Rattus norvegicus*) | Primary cultured neurons | ORIENT BIO | RGD Cat# 734476 RRID:RGD_734476) | SD rat |

## DNA constructs and antibodies

The plasmid encoding c-myc epitope-tagged mGlu7 in the N-terminus was described previously (*Suh et al., 2013*; *Suh et al., 2008*). FLAG-mGlu7 was generated by a site-directed fragment insertion protocol as previously described (*Park et al., 2016*; *Qi and Scholthof, 2008*). To insert FLAG sequence in the N-terminus, the oligonucleotides that contain partially complementary FLAG sequences (upper case letters) at the 5' end and mGlu7 sequences (lower case letters) at the 3' end were generated as follows: forward, 5'-TACAAAGACGATGACGACAAGacgcgtatgtacgcc-3'; reverse, 5'- GTCGTCATCGTCTTTGTAGTCtcgcgtggactgtgc-3'. The PCR reaction was performed using Phusion DNA polymerase (Thermo Fisher Scientific) according to the manufacturer's recommendations. HA-Ubiquitin (#17608), HA-Ubiquitin K48 (#17605), HA-Ubiquitin K63 (#17606), HA-Nedd4 (#27002), FLAG-β2AR (#14697), and β-arrestin 1-FLAG (#14687) were obtained from Addgene.

The antibodies used in this study were purchased from the following commercial sources: anti-mGlu7a (#07–239, EMD Millipore), anti-c-myc (9E10, Sigma Aldrich), anti-HA (16B12, Covance), anti-FLAG (Sigma Aldrich), anti-Ub (FK2, EMD Millipore; P4D1, Santa Cruz Biotechnology), anti-Nedd4 (#07–049, EMD Millipore; MAB6218, R&D Systems), anti-β-arrestin 1 (Abcam), anti-β-arrestin 2 (Cell Signaling Technology). Alexa Fluor 488 or 568-conjugated goat anti-mouse IgG, and horseradish peroxidase-conjugated goat anti-mouse or rabbit IgG were purchased from Thermo Fisher Scientific.

## Cell and primary neuron cultures

HEK 293 T cells were maintained in DMEM containing 10% fetal bovine serum (HyClone) and 1% L-glutamine (Invitrogen). Primary hippocampal or cortical neurons were prepared from E18 Sprague-Dawley rats (Orient Bio, Seongnam, Korea) following the guidelines of the Seoul National University

Institutional Animal Care and Use Committee (protocol No. SNU-161222-2-2). Briefly, hippocampi or cortices from rat embryos were isolated and incubated in chopping solution [Hanks' Balanced Salt Solution (Invitrogen), 10 mM HEPES (Sigma Aldrich), 0.11 mg/mL Deoxyribonuclease I (Sigma Aldrich), penicillin-streptomycin (Sigma Aldrich)] with 0.05% trypsin (Sigma Aldrich) for 12 min at 37° C. Hippocampi or cortices were washed three times with a cold chopping solution, and triturated 10–15 times with a fire-polished Pasteur pipette. The dissociated neurons were plated on poly-D-lysine-coated dishes in serum-free Neurobasal media (Invitrogen) with B-27 supplement and L-glutamine, and maintained at 37°C in a humidified 5% $CO_2$ incubator. Fresh medium was added every 2–3 days.

## Western blotting, immunoprecipitation, and surface biotinylation assay

HEK 293 T cells or cultured neurons were washed three times with ice-cold 1 X PBS and solubilized in TN lysis buffer [50 mM Tris-HCl, pH 8.0, 150 mM NaCl, 1% Triton X-100, and 0.1% sodium dodecyl sulfate (SDS)] with protease inhibitor cocktails (Roche) and 20 mM N-Ethylmaleimide (NEM, Sigma Aldrich). The insoluble materials were removed by centrifugation at $20,000 \times g$ for 15 min at 4°C. The supernatants were mixed with 6 X Laemmli sample buffer, boiled at 95°C for 5 min or incubated at 37°C for 20 min and subsequently at 80°C for 3 min. The samples were resolved with SDS-PAGE, and transferred to PVDF membranes. The membranes were blocked with 5% non-fat skim milk in 1 X TBST (0.1% Tween 20) for 1 hr at room temperature and then incubated with primary antibodies overnight at 4°C. Following several washings, the membranes were incubated with HRP-conjugated secondary antibodies. The immunoreactive bands were detected with SuperSignal West Pico Chemiluminescent Substrate (Thermo Fisher Scientific). For immunoprecipitation experiments, the lysates were pre-cleared with Sepharose 4B beads for 1 hr at 4°C. The pre-cleared supernatants were incubated with protein A- or G-Sepharose 4B beads (Sigma Aldrich or Amersham) and with primary antibodies for 2–4 hr at 4°C. The immunoprecipitates were then washed four times with lysis buffer and subjected to western blot analysis.

For the surface biotinylation assay, HEK 293 T cells or neurons were incubated with 0.5 mg/ml EZ-Link Sulfo-NHS-SS-biotin (Pierce) in 1 X PBS with 1 mM $MgCl_2$ and 0.1 mM $CaCl_2$ for 20 min at 4° C with gentle agitation. Remaining biotin was quenched by 50 mM glycine in 1 X PBS with 1 mM $MgCl_2$ and 0.1 mM $CaCl_2$. The biotinylated samples were lysed and supernatant was then incubated with 20 µl of Neutravidin or Streptavidin-agarose beads (Pierce) for 2 hr at 4°C. After washing the beads four times with lysis buffer, the bound proteins were analyzed by western blotting.

## Receptor internalization assay

The receptor internalization assay was performed as described elsewhere (*Choi et al., 2016*). Briefly, transfected neurons were incubated with 2 µg/ml anti-c-myc antibody for 10 min at room temperature to label surface-expressed receptors. The neurons were returned to the conditioned media containing 400 µM L-AP4 or vehicle at 37°C for 15 min to allow for the internalization of receptor-antibody complex. The neurons were fixed with 4% paraformaldehyde/4% sucrose in 1 X PBS for 20 min and blocked with 10% normal goat serum (NGS) for 1 hr. Surface receptors were incubated with Alexa Fluor 568 goat anti-mouse IgG. The neurons were then permeabilized with 0.25% Triton X-100 for 5 min and blocked with 10% NGS for 1 hr. The internalized receptors were labeled with Alexa Fluor 488 goat anti-mouse IgG. The neurons were washed and mounted with ProLong Antifade Kit (Invitrogen). Z-stacked maximum projection images were obtained using a confocal laser microscope (Nikon A1). The amount of internalization was quantified using MetaMorph software (Universal Imaging Corp.) as per manufacturer's instructions. The acquired images were separated into red (surface population remaining after internalization) and green (internalized population) channels. A region of interest (ROI) per single neuron was selected along with neurites in red channel, and the same ROI was copied to the green channel. The soma regions were excluded from ROI due to the saturation of signal intensity. The threshold was set to remove background and saturated signals from both channels. The integrated intensities (signal intensity summed over all pixels in the ROI) from each channel were obtained from the selected ROIs. The internalization rate was quantified as a ratio of green channel signal to total signals (red + green channel signals). The relative internalization rate was calculated as a ratio compared to the Ctl internalization rate.

## Virus production

For lentivirus production, FHUGW lentiviral vector was utilized. To knockdown endogenous gene expression, small hairpin RNA (shRNA) sequences were cloned under H1 promoter. Target shRNA sequences are as follows: rat Nedd4 (5'-GCCACAAATCAAGAGTTAA-3') (*Schwarz et al., 2010*), rat β-arrestin 1 (5'-AGCCTTCTGTGCTGAGAAC-3') (*Simard et al., 2013*), human β-arrestin 1 (5'-AAAGCCTTCTGCGCGGAGAAT-3') (*Shenoy et al., 2008*), rat β-arrestin 2 (5'-GGACCGGAAAGTG TTTGTG-3') (*Molteni et al., 2009*), human β-arrestin 2 (5'-GGCTTGTCCTTCCGCAAAG-3') (*Skånland et al., 2009*), control non-related target knockdown (5'-AGTGGATTCGAGAGCGTGT-3') (GenScript). For overexpression of Nedd4 WT, Nedd4 C867S, mGlu7 WT, or mGlu7 mutants in neurons, each cDNA was cloned under ubiquitin promoter after removal of EGFP in the FHUGW lentiviral vector. To produce lentiviral particles, HEK 293 T cells were co-transfected with lentiviral vector, Δ 8.9, and VSVG using X-tremeGENE (Roche). The medium was replaced by Neurobasal medium supplemented with 1% L-glutamine and Insulin-Transferrin-Sodium selenite (Sigma Aldrich). Supernatants containing the viral particles were collected 60 hr after transfection.

## Statistical analysis

Data are presented as the mean and the standard error of mean (S.E.M) based on three or more independent experiments. A Student's paired *t*-test or one-way ANOVA followed by Bonferroni's post-hoc test was used to compare the values of means. A *P* value of < 0.05 was considered statistically significant.

## Data availability

All data generated or analysed during this study are included in the manuscript and supporting files. Source data files have been provided for *Figures 1*, *2*, *3*, *5*, *6* and *7*.

## Acknowledgements

This work was supported by the National Research Foundation of Korea (NRF) grants (NRF-2016R1D1A1B03930951, NRF-2017M3C7A1029611, NRF-2018R1A2B6004759), the Korea Health Industry Development Institute (KHIDI) grant (HI18C0789), the SNUH Research Fund (0320150260), the Cooperative Research Program from SNUCM (800–20180195), and the Brain Korea 21 PLUS program.

## Additional information

### Funding

| Funder | Grant reference number | Author |
| --- | --- | --- |
| National Research Foundation | NRF-2016R1D1A1B03930951 | Sanghyeon Lee<br>Sunha Park<br>Hyojin Lee<br>Seulki Han<br>Jae-man Song<br>Young Ho Suh |
| National Research Foundation of Korea | NRF-2017M3C7A1029611 | Sanghyeon Lee<br>Sunha Park<br>Hyojin Lee<br>Seulki Han<br>Jae-man Song<br>Young Ho Suh |
| National Research Foundation of Korea | 2018R1A2B6004759 | Sanghyeon Lee<br>Sunha Park<br>Jae-man Song<br>Young Ho Suh |
| Korea Health Industry Development Institute | HI18C0789 | Sanghyeon Lee<br>Sunha Park<br>Jae-man Song<br>Young Ho Suh |

| | | |
|---|---|---|
| Seoul National University Hospital | 0320150260 | Sanghyeon Lee<br>Sunha Park<br>Hyojin Lee<br>Seulki Han<br>Jae-man Song<br>Young Ho Suh |
| Seoul National University | 800-20180195 | Sanghyeon Lee<br>Sunha Park<br>Jae-man Song<br>Young Ho Suh |
| Brain Korea 21 PLUS Program | | Sanghyeon Lee<br>Sunha Park<br>Hyojin Lee<br>Seulki Han<br>Jae-man Song<br>Young Ho Suh |

The funders had no role in study design, data collection and interpretation, or the decision to submit the work for publication.

## Author contributions

Sanghyeon Lee, Resources, Investigation, Methodology; Sunha Park, Hyojin Lee, Seulki Han, Jae-man Song, Investigation; Dohyun Han, Resources; Young Ho Suh, Conceptualization, Resources, Supervision, Funding acquisition, Methodology, Writing—original draft, Writing—review and editing

## Author ORCIDs

Sanghyeon Lee (iD) https://orcid.org/0000-0002-4456-3505
Young Ho Suh (iD) https://orcid.org/0000-0003-3979-1615

## Ethics

Animal experimentation: This study was performed in strict accordance with the recommendations in the Guide for the Care and Use of Laboratory Animals of the Seoul National University. All of the animals were handled according to approved protocol under the guidelines of the Seoul National University Institutional Animal Care and Use Committees (Approval Number: SNU-161222-2-2). The animals were sacrificed by $CO_2$ asphyxiation, and every effort was made to minimize suffering.

## Decision letter and Author response

Decision letter https://doi.org/10.7554/eLife.44502.027
Author response https://doi.org/10.7554/eLife.44502.028

# Additional files

## Supplementary files

• Transparent reporting form
DOI: https://doi.org/10.7554/eLife.44502.025

## Data availability

All data generated or analysed during this study are included in the manuscript and supporting files. Source data files have been provided for Figures 1, 2, 3, 5, 6, and 7.

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
