## [Decision Letter]

Thank you for submitting your article "Nedd4 E3 ligase and Beta-arrestins regulate ubiquitination, trafficking, and stability of the mGlu7 receptor" for consideration by *eLife*. Your article has been reviewed by three peer reviewers, one of whom is a member of our Board of Reviewing Editors, and the evaluation has been overseen by a Reviewing Editor and Richard Aldrich as the Senior Editor. The following individuals involved in review of your submission have agreed to reveal their identity: Nathalie Sans (Reviewer #3).

The reviewers have discussed the reviews with one another and the Reviewing Editor has drafted this decision to help you prepare a revised submission.

Summary:

This study examines the roles Nedd4 E3 ubiquitin ligase and β-arrestins in regulating mGlu7 metabotropic receptor ubiquitination, trafficking and signaling. Using cell biological and biochemical approaches in heterologous cells and neurons, the authors demonstrate that modulating or interfering with the expression of Nedd4 or β-arrestins alters the internalization of mGlu7 and the agonist-induced MAPK signaling in neurons. The main findings provides a model in which Nedd4 and β-arrestin act together as a complex to regulate mGlu7 surface expression and in turn, modulate presynaptic function where mGlu7 is localized. The experiments have been mostly well executed, and the manuscript is written clearly. All three reviewers have acknowledged the importance of the overall findings. However, they have also pointed out some shortcomings with the present dataset in fully supporting the conclusions. For instance, many of the experiments exploring the properties of mGlu7, Nedd4 and β-arrestin interactions have been performed only in HEK293 cells. How mGlu7 in its native environment in neurons might behave remains to be further clarified, including the formation of the complex of mGlu7, Nedd4 and β-arrestin. Moreover, a conclusive demonstration that the effects of manipulating the expression of Nedd4 or β-arrestins on mGlu7 trafficking and signaling is mediated by mGlu7 ubiquitination is lacking. Altogether, the following major points require careful consideration.

Essential revisions:

1) The design of biotinylation experiments shown in Figure 1D needs reconsideration. The logic of stimulating the neurons prior to surface biotinylation does not make sense. It should be done the other way around, so that all surface receptors are biotinylated prior to stimulation.

2) The precise roles for Nedd4 and β-arrestins on mGlu7 turnover, including (a) the requirement for β-arrestin-1 and -2 in Nedd4 recruitment to mGlu7 and (b) the extent involvement of proteosomal and lysosomal degradation pathways for mGlu7, should be established in neurons.

3) The requirement for the direct interaction between mGlu7, β-arrestin and Nedd4 in neurons should be established directly by taking advantage of the mGlu7 C-terminal tail and the intracellular loop ubiquitination site mutants.

4) Do the different mutated mGlu7 constructs have different surface expression levels? What about their stability?

5) Please clarify the basis by which Nedd4 was selected and tested. Are there additional E3 ubiquitin ligases that can regulate mGlu7, and could the authors explain Figure 1F data that show some remaining level of mGlu7 ubiquitinylation despite the loss of Nedd4.

6) In Figure 1C or 1D, in addition to ubiquitin that is used for IP, mGlu7 IP should also be tested, and a negative control for Western blot should be included. Also, to show that Nedd4 and β-arrestin act together as a complex to regulate mGlu7, the authors need to show that the three proteins are present in the same complex.

7) The images of surface receptor expression in neurons are not convincing at all. The authors need better quality images for quantification. Why does one not see the typical punctate staining for surface expression of receptors on those images? The details of image analysis need to be included in the method section.

[Editors' note: further revisions were requested prior to acceptance, as described below.]

Thank you for resubmitting your work entitled "Nedd4 E3 ligase and β-arrestins regulate ubiquitination, trafficking, and stability of the mGlu7 receptor" for further consideration at *eLife*. Your revised article has been favorably evaluated by Eve Marder (Senior Editor), a Reviewing Editor, and two reviewers.

While the manuscript has been improved, some new experiments in neurons appear to show differences from HEK cells. Moreover, there are a number of remaining issues needing consideration, one of which (point 3) requires an additional experiment, as outlined below:

1) Figure 1D. Please show quantification.

2) Binding affinity of Nedd4 or β-arrestin 1 to different mGlu7 ubiquitination site is not altered in neurons. This suggests that ubiquitination of mGlu7 is not a prerequisite for the complex formation, and this point needs to be discussed.

3) Figure 3K-N. Panels K,L: The new experiment examining the effects of knocking-down β-arrestin 1 or β-arrestin 2 on Nedd4 and mGlu7 interaction in neurons is lacking controls. Because in HEK293 cells, either β-arrestin 1 or β-arrestin 2 knock-down alone without the agonist tends to promote the binding between Nedd4 and mGlu7, it is crucial to include controls without the agonist in β-arrestin 1, β-arrestin 2, and β-arrestin 1/2 knock-down conditions as a reference. As it stands (without these controls) the data do not support the requirement for β-arrestins in the agonist-dependent binding of mGlu7 and Nedd4 (also in the Discussion section). Similarly, for Figure 3M and 3N, it is necessary to include non-stimulated controls to assess the levels of mGluR7 ubiquitination under basal conditions when β-arrestin1/2 are knocked-down.

---

## [Author Response]

Summary:This study examines the roles Nedd4 E3 ubiquitin ligase and β-arrestins in regulating mGlu7 metabotropic receptor ubiquitination, trafficking and signaling. […] Altogether, the following major points require careful consideration.

We are pleased that all reviewers have acknowledged the importance of the overall findings of our manuscript. We appreciate the detailed and constructive suggestions offered by the reviewers. We have thoroughly modified the manuscript and have included the results from new experiments.

We have provided a point-by-point response to the concerns of the reviewers below.

Essential revisions:1) The design of biotinylation experiments shown in Figure 1D needs reconsideration. The logic of stimulating the neurons prior to surface biotinylation does not make sense. It should be done the other way around, so that all surface receptors are biotinylated prior to stimulation.

We agree with the reviewer’s comment that biotinylation of surface receptors prior to stimulation will be a more appropriate way to evaluate ubiquitination of surface-expressed mGlu7 vs. intracellular mGlu7. This is because surface-expressed receptors can be internalized during the stimulation of neurons by the previous method. We have now performed the experiments as per the reviewer’s comment and found a similar yet more prominent result that the surface-expressed mGlu7 is ubiquitinated. We have replaced the previous Figure 1D with the new result in the revised manuscript. We appreciate the reviewer’s constructive suggestion.

2) The precise roles for Nedd4 and β-arrestins on mGlu7 turnover, including (a) the requirement for β-arrestin-1 and -2 in Nedd4 recruitment to mGlu7 and (b) the extent involvement of proteosomal and lysosomal degradation pathways for mGlu7, should be established in neurons.

To examine the requirement for β-arrestin 1 and 2 in Nedd4 recruitment to mGlu7 in neurons, we generated lentiviruses harboring β-arrestin 1 or 2 shRNA under H1 promoter. When β-arrestin 1 or 2 was knocked down in primary cortical neurons, agonist-induced Nedd4 recruitment to mGlu7 was not impaired. However, when both β-arrestin 1 and 2 were simultaneously knocked down, we observed a significant reduction of agonist-induced Nedd4 recruitment to mGlu7 (revised Figure 3K and 3L). This new finding is somewhat inconsistent with the result observed in heterologous cells in the original manuscript. We previously observed that either β-arrestin 1 or 2 knockdown (KD) efficiently reduced agonist-induced Nedd4 recruitment to mGlu7 in heterologous cells (Figure 3I and 3J in the revised manuscript). However, in neurons, we have now found that only the double KD of β-arrestin 1 and 2 abolishes agonist-induced Nedd4 binding to mGlu7 (revised Figure 3K and 3L). Endogenously expressed levels of β-arrestins are high in the brain, but relatively low in HEK 293T cells because only mGlu7 and Nedd4 were exogenously overexpressed in HEK 293T cells (Attramadal et al., 1992; Barlic et al., 1999). Therefore, we hypothesize that β-arrestin 1 and 2 may compensate for the function of each other in neurons due to their high expression but not in heterologous cells due to their stoichiometrically low expression.

Although both β-arrestin 1 and 2 KD were required to diminish agonist-induced Nedd4 recruitment to mGlu7 in neurons, agonist-induced mGlu7 ubiquitination was reduced by either the loss of β-arrestin 1 or 2 (Figure 3K–3N in the revised manuscript). It is unclear why the requirement for β-arrestins is different between Nedd4 recruitment to mGlu7 and agonist-induced mGlu7 ubiquitination. However, we postulate that β-arrestins may recruit isoform-specific ubiquitin E3 ligases other than Nedd4 in neurons.

We have included new data in Figures 3K and 3L, and the corresponding discussions in the Discussion section of the revised manuscript.

In addition, to determine the involvement of Nedd4 in mGlu7 stability in neurons, Nedd4 WT or Nedd4 C867S was expressed on cultured cortical neurons via lentivirus. We have now found that Nedd4 WT significantly reduces mGlu7 expression in neurons, whereas Nedd4 C867S does not. In addition, Nedd4 WT-mediated mGlu7 degradation is blocked by MG132 or leupeptin (revised Figure 7E and 7F). Thus, Nedd4-mediated ubiquitination is required for mGlu7 degradation, which is accomplished through the proteasomal and lysosomal degradation pathways in neurons.

We have included these new data in Figure 7E and 7F in the revised manuscript.

3) The requirement for the direct interaction between mGlu7, β-arrestin and Nedd4 in neurons should be established directly by taking advantage of the mGlu7 C-terminal tail and the intracellular loop ubiquitination site mutants.

In the Figure 1G of the original manuscript, we provided evidence that endogenous mGlu7 and Nedd4 interact together in cortical neurons, and their binding is increased by agonist treatment. To investigate if endogenous mGlu7, Nedd4, and β-arrestin 1 are present at the same complex, we performed co-immunoprecipitation assay using anti-β-arrestin 1 antibody in cortical neurons. We have found that three proteins are present in the same complex and that their binding affinity is increased by agonist treatment (revised Figure 3G and 3H).

In addition, to test direct interactions among mGlu7 ubiquitination site mutants, Nedd4, and β-arrestin 1, we expressed mGlu7 WT, iL 4K4R, Ct 8K8R, or 12K12R on cultured cortical neurons via lentivirus-mediated transduction and performed co-immunoprecipitation assay using anti-myc antibody. Although we observed the three proteins in the same complex, we found that binding affinity of Nedd4 or β-arrestin 1 to different mGlu7 ubiquitination site mutants was not affected by the Lys to Arg mutations in mGlu7 (revised Figure 3—figure supplement 2). A similar result was previously observed in beta2-AR, in which the ability of the 0-Kβ2AR (lysine-less mutant) to interact with β-arrestin was not altered by the Lys to Arg mutations in beta2-AR (Figure 8A in Shenoy et al., 2001). Therefore, on the same grounds, we argue that the interaction among mGlu7, Nedd4, and β-arrestins appears to be unaffected by the mGlu7 mutations.

4) Do the different mutated mGlu7 constructs have different surface expression levels? What about their stability?

We examined total and surface expression levels of mGlu7 WT, iL 4K4R, Ct 8K8R, 12K12R, Δ857, and Δ857/iL K688/689R mutants by surface biotinylation assay in cortical neurons following lentivirus-mediated transduction. We have observed that the relative surface expression (surface to total ratio) is increased in mGlu7 iL 4K4R, Δ857, and Δ857/iL K688/689R mutants. On the contrary, 8K8R or 12K12R mutant with mutations in mGlu7 cytoplasmic tail did not show an increase in relative surface expression (Author response image 1). The surface expression of different mGlu7 mutants can be complicated and unpredictable because of the following three reasons. First, because we were not able to specify any Lys residues essential for ubiquitination among the twelve Lys residues in mGlu7, post-translational modifications (PTMs) other than ubiquitination such as SUMOylation, acetylation, and neddylation can occur in the same Lys residues. Since different PTMs can affect surface expression of mGlu7 in opposite ways, it is difficult to evaluate surface expression of mGlu7 by ubiquitin modification solely with mGlu7 lysine mutants. For example, mGlu7 K889 residue can be SUMOylated and the deSUMOylated mGlu7 K889R mutant is robustly internalized (Choi et al., 2016). Second, the extensive point mutations in the cytoplasmic tail can alter the protein folding and conformation of mGlu7 and subsequently affect protein-protein interactions. Several interacting proteins and kinase/phosphatase that bind to the cytoplasmic tail of mGlu7 can regulate mGlu7 trafficking in the opposite ways (Suh et al., 2018). As the binding of interacting proteins and kinase/phosphatase can be altered in mGlu7 tail mutants, it is difficult to evaluate the effects of trafficking solely by ubiquitin modification. Third, different mutation constructs may possess different expression efficiency and protein kinetics. These may cause difficulties in evaluating surface expression and stability by a single timepoint experiment.

**Author response image 1. respfig1:** Surface expression levels of mGlu7 ubiquitination site mutants by cell surface biotinylation assay. mGlu7 WT, iL 4K4R, Ct 8K8R, 12K12R, Δ857, or Δ857/iL K688/689R mutants were expressed in cultured cortical neurons for 7 days via lentivirus-mediated transduction. At DIV14, after cell surface biotinylation with membrane impermeable Sulfo-NHS-LC-Biotin, surface proteins were isolated by Streptavidin-agarose beads. Western blotting was performed with the indicated antibodies.

Thus, we feel that this finding is not central to the conclusions of the study and is more confusing than illuminating. Rather, we would like to present two alternative approaches to assess the effects of ubiquitination on surface expression of mGlu7. First, we utilized Nedd4 WT and Nedd4 C867S, an E3 ubiquitin ligase mutant, to predict the effect of Nedd4-mediated ubiquitin modification on surface expression of mGlu7 as shown in Figure 5A and 5B in the revised manuscript. Next, to determine if ubiquitin modification regulates endocytosis of endogenous mGlu7, we have now utilized MLN7243, a cell-permeable small-molecule inhibitor of ubiquitin-activating enzyme (UAE, also known as E1 enzyme). When we treated cultured cortical neurons with MLN7243, we found that surface expression of endogenous mGlu7 is increased while mGlu7 ubiquitination is reduced (revised Figure 5–figure supplement 1). Thus, we suggest that Nedd4-mediated ubiquitination induces mGlu7 internalization from the surface plasma membrane of neurons.

5) Please clarify the basis by which Nedd4 was selected and tested. Are there additional E3 ubiquitin ligases that can regulate mGlu7, and could the authors explain Figure 1F data that show some remaining level of mGlu7 ubiquitinylation despite the loss of Nedd4.

At the beginning of our study, we observed that K63-linked ubiquitination is involved in mGlu7 ubiquitination. Hence, we searched for E3 ligase candidates that mediate K63-linked ubiquitination and are known to be expressed in neuronal synapses. Nedd4 and Mdm2 E3 ligases were the best candidates to meet these conditions. We tested these ligases and found that Nedd4 but not Mdm2 is involved in mGlu7 ubiquitination.

We absolutely agree with the possibility that other E3 ligases for mGlu7 ubiquitination may be present. Nedd4-2, a close relative of Nedd4, may be a good candidate. We have a plan to screen a number of E3 ubiquitin ligases and deubiquitinating enzymes (DUBs) in subsequent work.

We have included the reviewer’s comment in Discussion section of the revised manuscript as follows: “Of note, we observed some remaining levels of mGlu7 ubiquitination despite the loss of Nedd4 in Figure 1F. This result suggests that other E3 ligases than Nedd4 may play a role in mGlu7 ubiquitination in neurons.”

6) In Figure 1C or 1D, in addition to ubiquitin that is used for IP, mGlu7 IP should also be tested, and a negative control for Western blot should be included. Also, to show that Nedd4 and β-arrestin act together as a complex to regulate mGlu7, the authors need to show that the three proteins are present in the same complex.

Conversely, we have now examined the ubiquitination of mGlu7 following mGlu7 IP. GFP IP was used as a negative control. We have detected ubiquitinated bands of endogenous mGlu7 and included this result in revised Figure 1–figure supplement 1. In addition, we have added a result that shows that endogenous mGlu7, Nedd4, and β-arrestin 1 are present in the same complex in cortical neurons (revised Figure 3G and 3H). This was done via co-IP assay using rabbit anti-β-arrestin 1 antibody, or rabbit anti-GFP antibody as a negative control.

We have included these data in Figure 1–figure supplement 1, and Figure 3G and 3H in the revised manuscript.

7) The images of surface receptor expression in neurons are not convincing at all. The authors need better quality images for quantification. Why does one not see the typical punctate staining for surface expression of receptors on those images? The details of image analysis need to be included in the method section.

We have now replaced some old images in Figure 5 with new ones (Figure 5A, Figure 5E vec, and Figure 5E β-arr1) in the revised manuscript. The image quality deteriorated while converting the figure to DOC file. We have now improved resolution of images and presented better quality images in the revised Figure 5. We are sorry for this inconvenience. In addition, because of the lack of good mGlu7 antibodies that can recognize the extracellular N-terminus of mGlu7 and label surface-expressed endogenous mGlu7, we utilized anti-myc epitope antibody following the overexpression of extracellular myc epitope-tagged mGlu7 in neurons. For this reason, a typical punctate staining pattern may be not observed. We believe that this is the limit of our antibody uptake receptor internalization assay.

The amount of internalization was quantified using MetaMorph software as per manufacturer’s instructions. Z-stacked maximum projection images were obtained, and the images were separated into red (surface population remaining after internalization) and green (internalized population) channels. A region of interest (ROI) per single neuron was selected along with neurites in red channel, and the same ROI was copied to the green channel. The soma regions were excluded from ROI due to the saturation of signal intensity. The threshold was set to remove background and saturated signals from both channels. The integrated intensities (signal intensity summed over all pixels in the ROI) from each channel were obtained from the selected ROIs. The internalization rate was quantified as a ratio of green channel signal to total signals (red + green channel signals). The relative internalization rate was calculated as a ratio compared to the WT internalization rate. We have included these details of image analysis in the Materials and methods section in the revised manuscript.

[Editors' note: further revisions were requested prior to acceptance, as described below.]

1) Figure 1D. Please show quantification.

We have included quantification data of Figure 1D in the revised manuscript. Band intensities of ubiquitinated mGlu7 are normalized to total mGlu7 levels of each fraction and are shown as a ratio to the non-stimulated control of the surface fraction.

2) Binding affinity of Nedd4 or β-arrestin 1 to different mGlu7 ubiquitination site is not altered in neurons. This suggests that ubiquitination of mGlu7 is not a prerequisite for the complex formation, and this point needs to be discussed.

We have included the following sentences in Discussion section in the revised manuscript: “Although we observed β-arrestins and Nedd4 bind constitutively to mGlu7, binding affinity of Nedd4 or β-arrestin 1 to mGlu7 is not affected by ubiquitination site mutations in neurons (Figure 3—figure supplement 2). This result suggests that mGlu7 ubiquitination per se is not a prerequisite for the complex formation.”

3) Figure 3K-N. Panels K,L: The new experiment examining the effects of knocking-down β-arrestin 1 or β-arrestin 2 on Nedd4 and mGlu7 interaction in neurons is lacking controls. Because in HEK293 cells, either β-arrestin 1 or β-arrestin 2 knock-down alone without the agonist tends to promote the binding between Nedd4 and mGlu7, it is crucial to include controls without the agonist in β-arrestin 1, β-arrestin 2, and β-arrestin 1/2 knock-down conditions as a reference. As it stands (without these controls) the data do not support the requirement for β-arrestins in the agonist-dependent binding of mGlu7 and Nedd4 (also in the Discussion section). Similarly, for Figure 3M and 3N, it is necessary to include non-stimulated controls to assess the levels of mGluR7 ubiquitination under basal conditions when β-arrestin1/2 are knocked-down.

In our HEK 293T cell experiments, we were NOT able to observe that binding affinity of Nedd4 to mGlu7 is constitutively increased by either β-arrestin 1 knockdown (KD) or β-arrestin 2 KD in the absence of agonist stimulation. There was no significant difference between their bindings without the agonist. To clarify this point, we have replaced Figure 3J with a graph showing the statistical difference of each lane (revised Figure 3J).

To show basal (constitutive) interactions between Nedd4 and mGlu7 in the absence of agonist stimulation in neurons, we have performed co-immunoprecipitation assay including samples without the agonist in non-related target control KD and β-arrestin 1/2 KD conditions in neurons. We observed that in the absence of agonist stimulation, the interaction between Nedd4 and mGlu7 is not significantly affected by β-arrestin 1/2 KD, whereas in the presence of the agonist, their binding is increased in control KD but not by β-arrestin 1/2 KD (revised Figure 3—figure supplement 3). In addition, for Figure 3M and 3N, we have performed experiments including basal conditions in both control KD and β-arrestin1/2 KD. We have found that ubiquitination levels of mGlu7 are not different between control KD and β-arrestin1/2 KD in non-stimulated conditions, whereas mGlu7 ubiquitination is increased in control KD with the agonist but not by β-arrestin 1/2 KD (revise Figure 3—figure supplement 4).